# Novel Three-Way Decisions Models with Multi-Granulation Rough Intuitionistic Fuzzy Sets

**Zhan-Ao Xue [1,2,*], Dan-Jie Han [1,2,*], Min-Jie Lv [1,2] and Min Zhang [1,2]**

[1]   College of Computer and Information Engineering, Henan Normal University, Xinxiang 453007, China;
     lmj2921419592@163.com (M.-J.L.); zhang_min95@163.com (M.Z.)
[2]   Engineering Lab of Henan Province for Intelligence Business & Internet of Things,
     Henan Normal University, Xinxiang 453007, China
*   Correspondence: 121017@htu.edu.cn (Z.-A.X.); handanjie2017@163.com (D.-J.H.)

**Abstract:** The existing construction methods of granularity importance degree only consider the direct influence of single granularity on decision-making; however, they ignore the joint impact from other granularities when carrying out granularity selection. In this regard, we have the following improvements. First of all, we define a more reasonable granularity importance degree calculating method among multiple granularities to deal with the above problem and give a granularity reduction algorithm based on this method. Besides, this paper combines the reduction sets of optimistic and pessimistic multi-granulation rough sets with intuitionistic fuzzy sets, respectively, and their related properties are shown synchronously. Based on this, to further reduce the redundant objects in each granularity of reduction sets, four novel kinds of three-way decisions models with multi-granulation rough intuitionistic fuzzy sets are developed. Moreover, a series of concrete examples can demonstrate that these joint models not only can remove the redundant objects inside each granularity of the reduction sets, but also can generate much suitable granularity selection results using the designed comprehensive score function and comprehensive accuracy function of granularities.

**Keywords:** three-way decisions; intuitionistic fuzzy sets; multi-granulation rough intuitionistic fuzzy sets; granularity importance degree

## 1. Introduction

Pawlak [1,2] proposed rough sets theory in 1982 as a method of dealing with inaccuracy and uncertainty, and it has been developed into a variety of theories [3–6]. For example, the multi-granulation rough sets (MRS) model is one of the important developments [7,8]. The MRS can also be regarded as a mathematical framework to handle granular computing, which is proposed by Qian et al. [9]. Thereinto, the problem of granularity reduction is a vital research aspect of MRS. Considering the test cost problem of granularity structure selection in data mining and machine learning, Yang et al. constructed two reduction algorithms of cost-sensitive multi-granulation decision-making system based on the definition of approximate quality [10]. Through introducing the concept of distribution reduction [11] and taking the quality of approximate distribution as the measure in the multi-granulation decision rough sets model, Sang et al. proposed an $\alpha$-lower approximate distribution reduction algorithm based on multi-granulation decision rough sets, however, the interactions among multiple granularities were not considered [12]. In order to overcome the problem of updating reduction, when the large-scale data vary dynamically, Jing et al. developed an incremental attribute reduction approach based on knowledge granularity with a multi-granulation view [13]. Then other multi-granulation reduction methods have been put forward one after another [14–17].

The notion of intuitionistic fuzzy sets (IFS), proposed by Atanassov [18,19], was initially developed in the framework of fuzzy sets [20,21]. Within the previous literature, how to get reasonable membership and non-membership functions is a key issue. In the interest of dealing with fuzzy information better, many experts and scholars have expanded the IFS model. Huang et al. combined IFS with MRS to obtain intuitionistic fuzzy MRS [22]. On the basis of fuzzy rough sets, Liu et al. constructed covering-based multi-granulation fuzzy rough sets [23]. Moreover, multi-granulation rough intuitionistic fuzzy cut sets model was structured by Xue et al. [24]. In order to reduce the classification errors and the limitation of ordering by single theory, they further combined IFS with graded rough sets theory based on dominance relation and extended them to a multi-granulation perspective. [25]. Under the optimistic multi-granulation intuitionistic fuzzy rough sets, Wang et al. proposed a novel method to solve multiple criteria group decision-making problems [26]. However, the above studies rarely deal with the optimal granularity selection problem in intuitionistic fuzzy environments. The measure of similarity between intuitionistic fuzzy sets is also one of the hot areas of research for experts, and some similarity measures about IFS are summarized in references [27–29], whereas these metric formulas cannot measure the importance degree of multiple granularities in the same IFS.

For further explaining the semantics of decision-theoretic rough sets (DTRS), Yao proposed a three-way decisions theory [30,31], which vastly pushed the development of rough sets. As a risk decision-making method, the key strategy of three-way decisions is to divide the domain into acceptance, rejection, and non-commitment. Up to now, researchers have accumulated a vast literature on its theory and application. For instance, in order to narrow the applications limits of three-way decisions model in uncertainty environment, Zhai et al. extended the three-way decisions models to tolerance rough fuzzy sets and rough fuzzy sets, respectively, the target concepts are relatively extended to tolerance rough fuzzy sets and rough fuzzy sets [32,33]. To accommodate the situation where the objects or attributes in a multi-scale decision table are sequentially updated, Hao et al. used sequential three-way decisions to investigate the optimal scale selection problem [34]. Subsequently, Luo et al. applied three-way decisions theory to incomplete multi-scale information systems [35]. With respect to multiple attribute decision-making, Zhang et al. study the inclusion relations of neutrosophic sets in their case in reference [36]. For improving the classification correct rate of three-way decisions, Zhang et al. proposed a novel three-way decisions model with DTRS by considering the new risk measurement functions through the utility theory [37]. Yang et al. combined three-way decisions theory with IFS to obtain novel three-way decision rules [38]. At the same time, Liu et al. explored the intuitionistic fuzzy three-way decision theory based on intuitionistic fuzzy decision systems [39]. Nevertheless, Yang et al. [38] and Liu et al. [39] only considered the case of a single granularity, and did not analyze the decision-making situation of multiple granularities in an intuitionistic fuzzy environment. The DTRS and three-way decisions theory are both used to deal with decision-making problems, so it is also enlightening for us to study three-way decisions theory through DTRS. An extension version that can be used to multi-periods scenarios has been introduced by Liang et al. using intuitionistic fuzzy decision- theoretic rough sets [40]. Furthermore, they introduced the intuitionistic fuzzy point operator into DTRS [41]. The three-way decisions are also applied in multiple attribute group decision making [42], supplier selection problem [43], clustering analysis [44], cognitive computer [45], and so on. However, they have not applied the three-way decisions theory to the optimal granularity selection problem. To solve this problem, we have expanded the three-way decisions models.

The main contributions of this paper include four points:

(1) The new granularity importance degree calculating methods among multiple granularities (i.e., $sig'^{\Delta}_{in}(A_i, A\prime, D)$ and $sig'^{\Delta}_{out}(A_i, A\prime, D)$) are given respectively, which can generate more discriminative granularities.

(2) Optimistic optimistic multi-granulation rough intuitionistic fuzzy sets (OOMRIFS) model, optimistic pessimistic multi-granulation rough intuitionistic fuzzy sets (OIMRIFS) model,

pessimistic optimistic multi-granulation rough intuitionistic fuzzy sets (IOMRIFS) model and pessimistic pessimistic multi-granulation rough intuitionistic fuzzy sets (IIMRIFS) model are constructed by combining intuitionistic fuzzy sets with the reduction of the optimistic and pessimistic multi-granulation rough sets. These four models can reduce the subjective errors caused by a single intuitionistic fuzzy set.

(3) We put forward four kinds of three-way decisions models based on the proposed four multi-granulation rough intuitionistic fuzzy sets (MRIFS), which can further reduce the redundant objects in each granularity of reduction sets.

(4) Comprehensive score function and comprehensive accuracy function based on MRIFS are constructed. Based on this, we can obtain the optimal granularity selection results.

The rest of this paper is organized as follows. In Section 2, some basic concepts of MRS, IFS, and three-way decisions are briefly reviewed. In Section 3, we propose two new granularity importance degree calculating methods and a granularity reduction Algorithm 1. At the same time, a comparative example is given. Four novel MRIFS models are constructed in Section 4, and the properties of the four models are verified by Example 2. Section 5 proposes some novel three-way decisions models based on above four new MRIFS, and the comprehensive score function and comprehensive accuracy function based on MRIFS are built. At the same time, through Algorithm 2, we make the optimal granularity selection. In Section 6, we use Example 3 to study and illustrate the three-way decisions models based on new MRIFS. Section 7 concludes this paper.

## 2. Preliminaries

The basic notions of MRS, IFS, and three-way decisions theory are briefly reviewed in this section. Throughout the paper, we denote $U$ as a nonempty object set, i.e., the universe of discourse and $A = \{A_1, A_2, \cdots, A_m\}$ is an attribute set.

**Definition 1** ([9]). *Suppose IS* $=< U, A, V, f >$ *is a consistent information system, $A = \{A_1, A_2, \cdots, A_m\}$ is an attribute set. And $R_{A_i}$ is an equivalence relation generated by A. $[x]_{A_i}$ is the equivalence class of $R_{A_i}$, $\forall X \subseteq U$, the lower and upper approximations of optimistic multi-granulation rough sets (OMRS) of X are defined by the following two formulas:*

$$\sum_{i=1}^{m} A_i^{O}(X) = \{x \in U | [x]_{A_1} \subseteq X \vee [x]_{A_2} \subseteq X \vee [x]_{A_3} \subseteq X \ldots \vee [x]_{A_m} \subseteq X\};$$

$$\overline{\sum_{i=1}^{m} A_i}^{O}(X) = \sim (\underline{\sum_{i=1}^{m} A_i}^{O}(\sim X)).$$

*where $\vee$ is a disjunction operation, $\sim X$ is a complement of X, if $\underline{\sum_{i=1}^{m} A_i}^{O}(X) \neq \overline{\sum_{i=1}^{m} A_i}^{O}(X)$, the pair $(\underline{\sum_{i=1}^{m} A_i}^{O}(X), \overline{\sum_{i=1}^{m} A_i}^{O}(X))$ is referred to as an optimistic multi-granulation rough set of X.*

**Definition 2** ([9]). *Let IS $=< U, A, V, f >$ be an information system, where $A = \{A_1, A_2, \cdots, A_m\}$ is an attribute set, and $R_{A_i}$ is an equivalence relation generated by A. $[x]_{A_i}$ is the equivalence class of $R_{A_i}$, $\forall X \subseteq U$, the pessimistic multi-granulation rough sets (IMRS) of X with respect to A are defined as follows:*

$$\underline{\sum_{i=1}^{m} A_i}^{I}(X) = \{x \in U | [x]_{A_1} \subseteq X \wedge [x]_{A_2} \subseteq X \wedge [x]_{A_3} \subseteq X \wedge \ldots \wedge [x]_{A_m} \subseteq X\};$$

$$\overline{\sum_{i=1}^{m} A_i}^{I}(X) = \sim (\underline{\sum_{i=1}^{m} A_i}^{I}(\sim X)).$$

*where* $[x]_{A_i} (1 \leq i \leq m)$ *is equivalence class of x for* $A_i$, $\wedge$ *is a conjunction operation, if* $\sum\limits_{i=1}^{m} A_i^{I} (X) \neq$ $\overline{\sum\limits_{i=1}^{m} A_i}^{I} (X)$, *the pair* $(\sum\limits_{i=1}^{m} A_i^{I} (X), \overline{\sum\limits_{i=1}^{m} A_i}^{I} (X))$ *is referred to as a pessimistic multi-granulation rough set of X.*

**Definition 3** ([18,19]). *Let U be a finite non-empty universe set, then the IFS E in U are denoted by:*

$$E = \{< x, \mu_E(x), \nu_E(x) > | x \in U\},$$

*where* $\mu_E(x) : U \rightarrow [0,1]$ *and* $\nu_E(x) : U \rightarrow [0,1]$. $\mu_E(x)$ *and* $\nu_E(x)$ *are called membership and non-membership functions of the element x in E with* $0 \leq \mu_E(x) + \nu_E(x) \leq 1$. *For* $\forall x \in U$, *the hesitancy degree function is defined as* $\pi_E(x) = 1 - \mu_E(x) - \nu_E(x)$, *obviously,* $\pi_E(x) : U \rightarrow [0,1]$. *Suppose* $\forall E_1, E_2 \in IFS(U)$, *the basic operations of* $E_1$ *and* $E_2$ *are given as follows:*

(1)  $E_1 \subseteq E_2 \Leftrightarrow \mu_{E_1}(x) \leq \mu_{E_2}(x), \nu_{E_1}(x) \geq \nu_{E_2}(x), \forall x \in U$;
(2)  $A = B \Leftrightarrow \mu_A(x) = \mu_B(x), \nu_A(x) = \nu_B(x), \forall x \in U$;
(3)  $E_1 \cup E_2 = \{< x, \max\{\mu_{E_1}(x), \mu_{E_2}(x)\}, \min\{\nu_{E_1}(x), \nu_{E_2}(x)\} > | x \in U\}$;
(4)  (4) $E_1 \cap E_2 = \{< x, \min\{\mu_{E_1}(x), \mu_{E_2}(x)\}, \max\{\nu_{E_1}(x), \nu_{E_2}(x)\} > | x \in U\}$;
(5)  (5) $\sim E_1 = \{< x, \nu_{E_1}(x), \mu_{E_1}(x) > | x \in U\}$.

**Definition 4** ([30,31]). *Let* $U = \{x_1, x_2, \cdots, x_n\}$ *be a universe of discourse,* $\xi = \{\omega_P, \omega_N, \omega_B\}$ *represents the decisions of dividing an object x into receptive POS(X), rejective NEG(X), and boundary regions BND(X), respectively. The cost functions* $\lambda_{PP}$, $\lambda_{NP}$ *and* $\lambda_{BP}$ *are used to represent the three decision- making costs of* $\forall x \in U$, *and the cost functions* $\lambda_{PN}$, $\lambda_{NN}$ *and* $\lambda_{BN}$ *are used to represent the three decision-making costs of* $\forall x \notin U$, *as shown in Table 1.*

**Table 1.** Cost matrix of decision actions.

| Decision Actions | Decision Functions | |
| --- | --- | --- |
| | $X$ | $\sim X$ |
| $\omega_P$ | $\lambda_{PP}$ | $\lambda_{PN}$ |
| $\omega_B$ | $\lambda_{BP}$ | $\lambda_{BN}$ |
| $\omega_N$ | $\lambda_{NP}$ | $\lambda_{NN}$ |

According to the minimum-risk principle of Bayesian decision procedure, three-way decisions rules can be obtained as follows:

(P): If $P(X|[x]) \geq \alpha$, then $x \in POS(X)$;
(N): If $P(X|[x]) \leq \beta$, then $x \in NEG(X)$;
(B): If $\beta < P(X|[x]) < \alpha$, then $x \in BND(X)$.
Here $\alpha$, $\beta$ and $\gamma$ represent respectively:

$$\alpha = \frac{\lambda_{PN} - \lambda_{BN}}{(\lambda_{PN} - \lambda_{BN}) + (\lambda_{BP} - \lambda_{PP})};$$

$$\beta = \frac{\lambda_{BN} - \lambda_{NN}}{(\lambda_{BN} - \lambda_{NN}) + (\lambda_{NP} - \lambda_{BP})};$$

$$\gamma = \frac{\lambda_{PN} - \lambda_{NN}}{(\lambda_{PN} - \lambda_{NN}) + (\lambda_{NP} - \lambda_{PP})}.$$

### 3. Granularity Reduction Algorithm Derives from Granularity Importance Degree

**Definition 5** ([10,12]). *Let $DIS = (U, C \cup D, V, f)$ be a decision information system, $A = \{A_1, A_2, \cdots, A_m\}$ are m sub-attributes of condition attributes C. $U/D = \{X_1, X_2, \cdots, X_s\}$ is the partition induced by the decision attributes D, then approximation quality of $U/D$ about granularity set A is defined as:*

$$\gamma(A, D) = \frac{\left| \cup \left\{ \sum_{i=1}^{m} A_i^{\Delta}(X_t) | 1 \le t \le s \right\} \right|}{|U|}.$$

*where $|X|$ denotes the cardinal number of set X. $\Delta \in \{O, I\}$ represents two cases of optimistic and pessimistic multi-granulation rough sets, the same as the following.*

**Definition 6** ([12]). *Let $DIS = (U, C \cup D, V, f)$ be a decision information system, $A = \{A_1, A_2, \cdots, A_m\}$ are m sub-attributes of C, $A\prime \subseteq A$, $X \in U/D$,*

(1) *If $\sum\limits_{i=1, A_i \in A}^{m} A_i^{\Delta}(X) \ne \sum\limits_{i=1, A_i \in A-A\prime}^{m} A_i^{\Delta}(X)$, then A' is important in A for X;*

(2) *If $\sum\limits_{i=1, A_i \in A}^{m} A_i^{\Delta}(X) = \sum\limits_{i=1, A_i \in A-A\prime}^{m} A_i^{\Delta}(X)$, then A' is not important in A for X.*

**Definition 7** ([10,12]). *Suppose $DIS = (U, C \cup D, V, f)$ is a decision information system, $A = \{A_1, A_2, \cdots, A_m\}$ are m sub-attributes of C, $A\prime \subseteq A$. $\forall A_i \in A\prime$, on the granularity sets A', the internal importance degree of $A_i$ for D can be defined as follows:*

$$sig_{in}^{\Delta}(A_i, A\prime, D) = |\gamma(A\prime, D) - \gamma(A\prime - \{A_i\}, D)|.$$

**Definition 8** ([10,12]). *Let $DIS = (U, C \cup D, V, f)$ be a decision information system, $A = \{A_1, A_2, \cdots, A_m\}$ are m sub-attributes of C, $A\prime \subseteq A$. $\forall A_i \in A - A\prime$, on the granularity sets A', the external importance degree of $A_i$ for D can be defined as follows:*

$$sig_{out}^{\Delta}(A_i, A\prime, D) = |\gamma(A_i \cup A\prime, D) - \gamma(A\prime, D)|.$$

**Theorem 1.** *Let $DIS = (U, C \cup D, V, f)$ be a decision information system, $A = \{A_1, A_2, \cdots, A_m\}$ are m sub-attributes of C, $A\prime \subseteq A$.*

(1) *For $\forall A_i \in A\prime$, on the basis of attribute subset family A', the granularity importance degree of $A_i$ in A' with respect to D is expressed as follows:*

$$sig_{in}^{\Delta}(A_i, A\prime, D) = \frac{1}{m-1} \sum |sig_{in}^{\Delta}(\{A_k, A_i\}, A\prime, D) - sig_{in}^{\Delta}(A_k, A\prime - \{A_i\}, D)|.$$

*where $1 \le k \le m, k \ne i$, the same as the following.*

(2) *For $\forall A_i \in A - A\prime$, on the basis of attribute subset family A', the granularity importance degree of $A_i$ in $A - A\prime$ with respect to D, we have:*

$$sig_{out}^{\Delta}(A_i, A\prime, D) = \frac{1}{m-1} \sum |sig_{out}^{\Delta}(\{A_k, A_i\}, \{A_i\} \cup A\prime, D) - sig_{out}^{\Delta}(A_k, A\prime, D)|.$$

**Proof.** (1) According to Definition 7, then

$$
\begin{aligned}
sig_{in}^{\Delta}(A_i, A\prime, D) &= |\gamma(A\prime, D) - \gamma(A\prime - \{A_i\}, D)| \\
&= \tfrac{m-1}{m-1}|\gamma(A\prime, D) - \gamma(A\prime - \{A_i\}, D)| + \sum |\gamma(A\prime - \{A_k, A_i\}, D) - \gamma(A\prime - \{A_k, A_i\}, D)| \\
&= \tfrac{1}{m-1}\sum \left(|\gamma(A\prime, D) - \gamma(A\prime - \{A_k, A_i\}, D) - (\gamma(A\prime - \{A_i\}, D) - \gamma(A\prime - \{A_k, A_i\}, D)|\right) \\
&= \tfrac{1}{m-1}\sum |sig_{in}^{\Delta}(\{A_k, A_i\}, A\prime, D) - sig_{in}^{\Delta}(A_k, A\prime - \{A_i\}, D)|.
\end{aligned}
$$

(2) According to Definition 8, we can get:

$$
\begin{aligned}
sig_{out}^{\Delta}(A_i, A\prime, D) &= |\gamma(\{A_i\} \cup A\prime, D) - \gamma(A\prime, D)| \\
&= \tfrac{m-1}{m-1}|\gamma(\{A_i\} \cup A\prime, D) - \gamma(A\prime, D)| - \sum |\gamma(A\prime - \{A_k\}, D) - \gamma(A\prime - \{A_k\}, D)| \\
&= \tfrac{1}{m-1}\sum \left(|\gamma(\{A_i\} \cup A\prime, D) - \gamma(A\prime - \{A_k\}, D)| - |(\gamma(A\prime - \{A_k\}, D) - \gamma(A\prime, D)|\right) \\
&= \tfrac{1}{m-1}\sum |sig_{out}^{\Delta}(\{A_k, A_i\}, \{A_i\} \cup A\prime, D) - sig_{out}^{\Delta}(A_k, A\prime, D)|.
\end{aligned}
$$

□

In Definitions 7 and 8, only the direct effect of a single granularity on the whole granularity sets is given, without considering the indirect effect of the remaining granularities on decision-making. The following Definitions 9 and 10 synthetically analyze the interdependence between multiple granularities and present two new methods for calculating granularity importance degree.

**Definition 9.** *Let $DIS = (U, C \cup D, V, f)$ be a decision information system, $A = \{A_1, A_2, \cdots, A_m\}$ are m sub-attributes of C, $A\prime \subseteq A$. $\forall A_i, A_k \in A\prime$, on the attribute subset family, A, the new internal importance degree of $A_i$ relative to D is defined as follows:*

$$
sig\prime_{in}^{\Delta}(A_i, A\prime, D) = sig_{in}^{\Delta}(A_i, A\prime, D) + \frac{1}{m-1}\sum |sig_{in}^{\Delta}(A_k, A\prime - \{A_i\}, D) - sig_{in}^{\Delta}(A_k, A\prime, D)|.
$$

$sig_{in}^{\Delta}(A_i, A\prime, D)$ *and* $\frac{1}{m-1}\sum |sig_{in}^{\Delta}(A_k, A\prime - \{A_i\}, D) - sig_{in}^{\Delta}(A_k, A\prime, D)|$ *respectively indicate the direct and indirect effects of granularity $A_i$ on decision-making. When* $|sig_{in}^{\Delta}(A_k, A\prime - \{A_i\}, D) - sig_{in}^{\Delta}(A_k, A\prime, D)| > 0$ *is satisfied, it is shown that the granularity importance degree of $A_k$ is increased by the addition of $A_i$ in attribute subset $A\prime - \{A_i\}$, so the granularity importance degree of $A_k$ should be added to $A_i$. Therefore, when there are m sub-attributes, we should add* $\frac{1}{m-1}\sum |sig_{in}^{\Delta}(A_k, A\prime - \{A_i\}, D) - sig_{in}^{\Delta}(A_k, A\prime, D)|$ *to the granularity importance degree of $A_i$.*

*If* $|sig_{in}^{\Delta}(A_k, A\prime - \{A_i\}, D) - sig_{in}^{\Delta}(A_k, A\prime, D)| = 0$ *and $k \neq i$, then it shows that there is no interaction between granularity $A_i$ and other granularities, which means* $sig\prime_{in}^{\Delta}(A_i, A\prime, D) = sig_{in}^{\Delta}(A_i, A\prime, D)$.

**Definition 10.** *Let $DIS = (U, C \cup D, V, f)$ be a decision information system, $A = \{A_1, A_2, \cdots, A_m\}$ be m sub-attributes of C, $A\prime \subseteq A$. $\forall A_i \in A - A\prime$, the new external importance degree of $A_i$ relative to D is defined as follows:*

$$
sig\prime_{out}^{\Delta}(A_i, A\prime, D) = sig_{out}^{\Delta}(A_i, A\prime, D) + \frac{1}{m-1}\sum |sig_{out}^{\Delta}(A_k, A\prime, D) - sig_{out}^{\Delta}(A_k, \{A_i\} \cup A\prime, D)|.
$$

Similarly, the new external importance degree calculation formula has a similar effect.

**Theorem 2.** *Let $DIS = (U, C \cup D, V, f)$ be a decision information system, $A = \{A_1, A_2, \cdots, A_m\}$ be m sub-attributes of C, $A\prime \subseteq A$, $\forall A_i \in A\prime$. The improved internal importance can be rewritten as:*

$$
sig\prime_{in}^{\Delta}(A_i, A\prime, D) = \frac{1}{m-1}\sum sig_{in}^{\Delta}(A_i, A\prime - \{A_k\}, D).
$$

**Proof.**

$$
\begin{aligned}
sig'^{\Delta}_{in}(A_i, A\prime, D) \ &= sig^{\Delta}_{in}(A_i, A\prime, D) + \tfrac{1}{m-1}\sum |sig^{\Delta}_{in}(A_k, A\prime - \{A_i\}, D) - sig^{\Delta}_{in}(A_k, A\prime, D)| \\
&= \tfrac{m-1}{m-1}|\gamma(A\prime, D) - \gamma(A\prime - \{A_i\}, D)| + \tfrac{1}{m-1}\sum ||\gamma(A\prime - \{A_i\}, D) - \\
&\quad \gamma(A\prime - \{A_k, A_i\}, D)| - |\gamma(A\prime, D) - \gamma(A\prime - \{A_k\}, D)|| \\
&= \tfrac{1}{m-1}\sum |\gamma(A\prime - \{A_k\}, D) - \gamma(A\prime - \{A_k, A_i\}, D)| \\
&= \tfrac{1}{m-1}\sum sig^{\Delta}_{in}(A_i, A\prime - \{A_k\}, D).
\end{aligned}
$$

□

**Theorem 3.** *Let $DIS = (U, C \cup D, V, f)$ be a decision information system, $A = \{A_1, A_2, \cdots, A_m\}$ are m sub-attributes of C, $A\prime \subseteq A$. The improved external importance can be expressed as follows:*

$$
sig'^{\Delta}_{out}(A_i, A\prime, D) = \frac{1}{m-1}\sum sig^{\Delta}_{out}(A_i, \{A_k\} \cup A\prime, D).
$$

**Proof.**

$$
\begin{aligned}
sig'^{\Delta}_{out}(A_i, A\prime, D) \ &= sig^{\Delta}_{out}(A_i, A\prime, D) + \tfrac{1}{m-1}\sum |(sig^{\Delta}_{out}(A_k, A\prime, D) - sig^{\Delta}_{out}(A_k, \{A_i\} \cup A\prime, D))| \\
&= \tfrac{m-1}{m-1}|\gamma(\{A_i\} \cup A\prime, D) - \gamma(A\prime, D)| + \tfrac{1}{m-1}\sum ||\gamma(A\prime, D) - \gamma(\{A_k\} \cup A\prime, D)| - \\
&\quad |\gamma(\{A_i\} \cup A\prime, D)|| \\
&= \tfrac{1}{m-1}\sum |\gamma(\{A_i, A_k\} \cup A\prime, D) - \gamma(\{A_i\} \cup A\prime, D)| \\
&= \tfrac{1}{m-1}\sum sig^{\Delta}_{out}(A_i, \{A_k\} \cup A\prime, D).
\end{aligned}
$$

□

　　Theorems 2 and 3 show that when $sig^{\Delta}_{in}(A_i, A\prime - \{A_k\}, D) = 0$ $(sig^{\Delta}_{out}(A_i, \{A_k\} \cup A\prime, D) = 0)$ is satisfied, having $sig'^{\Delta}_{in}(A_i, A\prime, D) = 0$ $(sig'^{\Delta}_{out}(A_i, A\prime, D) = 0)$. And each granularity importance degree is calculated on the basis of removing $A_k$ from $A\prime$, which makes it more convenient for us to choose the required granularity.

　　According to [10,12], we can get optimistic and pessimistic multi-granulation lower approximations $L^O$ and $L^I$. The granularity reduction algorithm based on improved granularity importance degree is derived from Theorems 2 and 3, as shown in Algorithm 1.

---

**Algorithm 1.** Granularity reduction algorithm derives from granularity importance degree

---

**Input:** $DIS = (U, C \cup D, V, f)$, $A = \{A_1, A_2, \cdots, A_m\}$ are m sub-attributes of C, $A\prime \subseteq A, \forall A_i \in A\prime$,
$U/D = \{X_1, X_2, \cdots, X_s\}$;
**Output:** A granularity reduction set $A_i^\Delta$ of this information system.
1: set up $A_i^\Delta \leftarrow \phi, 1 \le h \le m$;
2: compute $U/D$, optimistic and pessimistic multi-granulation lower approximations $L^\Delta$;
3: **for** $\forall A_i \in A$
4:    compute $sig\prime_{in}^\Delta(A_i, A\prime, D)$ via Definition 9;
5:    **if** $(sig\prime_{in}^\Delta(A_i, A\prime, D) > 0)$ **then** $A_i^\Delta = A_i^\Delta \cup A_i$;
6:    **end**
7:    **for** $\forall A_i \in A - A_i^\Delta$
8:        **if** $\gamma(A_i^\Delta, D) = \gamma(A, D)$ **then** compute $sig_{out}^{\prime,\Delta}(A_i, A\prime, D)$ via Definition 10;
9:        **end**
10:       **if** $sig\prime_{out}^\Delta(A_h, A\prime, D) = \max\{sig\prime_{out}^\Delta(A_h, A\prime, D)\}$ **then** $A_i^\Delta = A_i^\Delta \cup A_h$;
11:       **end**
12:   **end**
13:   **for** $\forall A_i \in A_i^\Delta$,
14:       **if** $\gamma(A_i^\Delta - A_i, D) = \gamma(A, D)$ **then** $A_i^\Delta = A_i^\Delta - A_i$;
15:       **end**
16:   **end**
17: **return** granularity reduction set $A_i^\Delta$;
18: **end**

---

Therefore, we can obtain two reductions by utilizing Algorithm 1.

**Example 1.** *This paper calculates the granularity importance of 10 on-line investment schemes given in Reference [12]. After comparing and analyzing the obtained granularity importance degree, we can obtain the reduction results of 5 evaluation sites through Algorithm 1, and the detailed calculation steps are as follows.*

According to [12], we can get $A = \{A_1, A_2, A_3, A_4, A_5\}$, $A\prime \subseteq A$, $U/D = \{\{x_1, x_2, x_4, x_6, x_8\}, \{x_3, x_5, x_7, x_9, x_{10}\}\}$.

(1)    Reduction set of OMRS

First of all, we can calculate the internal importance degree of OMRS by Theorem 2 as shown in Table 2.

**Table 2.** Internal importance degree of optimistic multi-granulation rough sets (OMRS).

|  | $A_1$ | $A_2$ | $A_3$ | $A_4$ | $A_5$ |
|---|---|---|---|---|---|
| $sig_{in}^O(A_i, A\prime, D)$ | 0 | 0.15 | 0.05 | 0 | 0.05 |
| $sig\prime_{in}^O(A_i, A\prime, D)$ | 0.025 | 0.375 | 0.225 | 0 | 0 |

Then, according to Algorithm 1, we can deduce the initial granularity set is $\{A_1, A_2, A_3\}$. Inspired by Definition 5, we obtain $r^O(\{A_2, A_3\}, D) = r^O(A, D) = 1$. So, the reduction set of the OMRS is $A_i^O = \{A_2, A_3\}$.

As shown in Table 2, when using the new method to calculate internal importance degree, more discriminative granularities can be generated, which are more convenient for screening out the required granularities. In literature [12], the approximate quality of granularity $A_2$ in the reduction set is different from that of the whole granularity set, so it is necessary to calculate the external importance degree again. When calculating the internal and external importance degree, References [10,12] only considered the direct influence of the single granularity on the granularity $A_2$, so the influence of the granularity $A_2$ on the overall decision-making can't be fully reflected.

(2)  Reduction set of IMRS

Similarly, by using Theorem 2, we can get the internal importance degree of each site under IMRS, as shown in Table 3.

**Table 3.** Internal importance degree of pessimistic multi-granulation rough sets (IMRS).

|  | $A_1$ | $A_2$ | $A_3$ | $A_4$ | $A_5$ |
|---|---|---|---|---|---|
| $sig_{in}^{I}(A_i, A\prime, D)$ | 0 | 0.05 | 0 | 0 | 0 |
| $sig\prime_{in}^{I}(A_i, A\prime, D)$ | 0 | 0.025 | 0 | 0.025 | 0.025 |

According to Algorithm 1, the sites 2, 4, and 5 with internal importance degrees greater than 0, which are added to the granularity reduction set as the initial granularity set, and then the approximate quality of it can be calculated as follows:

$$r^I(\{A_2, A_4\}, D) = r^I(\{A_4, A_5\}, D) = r^I(A, D) = 0.2.$$

Namely, the reduction set of IMRS is $A_i^I = \{A_2, A_4\}$ or $A_i^I = \{A_4, A_5\}$ without calculating the external importance degree.

In this paper, when calculating the internal and external importance degree of each granularity, the influence of removing other granularities on decision-making is also considered. According to Theorem 2, after calculating the internal importance degree of OMRS and IMRS, if the approximate quality of each granularity in the reduction sets are the same as the overall granularities, it is not necessary to calculate the external importance degree again, which can reduce the amount of computation.

## 4. Novel Multi-Granulation Rough Intuitionistic Fuzzy Sets Models

In Example 1, two reduction sets are obtained under IMRS, so we need a novel method to obtain more accurate granularity reduction results by calculating granularity reduction.

In order to obtain the optimal determined site selection result, we combine the optimistic and pessimistic multi-granulation reduction sets based on Algorithm 1 with IFS, respectively, and construct the following four new MRIFS models.

**Definition 11** ([22,25]). *Suppose $IS = (U, A, V, f)$ is an information system, $A = \{A_1, A_2, \cdots, A_m\}$. $\forall E \subseteq U$, $E$ are IFS. Then the lower and upper approximations of optimistic MRIFS of $A_i$ are respectively defined by:*

$$\sum_{i=1}^{m} \underline{R_{A_i}}^{O} (E) = \{< x, \mu_{\sum_{i=1}^{m} R_{A_i}}^{O}(E)}(x), v_{\sum_{i=1}^{m} R_{A_i}}^{O}(E)}(x) > | x \in U\};$$

$$\overline{\sum_{i=1}^{m} R_{A_i}}^{O} (E) = \{< x, \mu_{\overline{\sum_{i=1}^{m} R_{A_i}}^{O}(E)}(x), v_{\overline{\sum_{i=1}^{m} R_{A_i}}^{O}(E)}(x) > | x \in U\}.$$

*where*

$$\mu_{\sum_{i=1}^{m} \underline{R_{A_i}}^{O}(E)}(x) = \overset{m}{\underset{i=1}{\vee}} \inf_{y \in [x]_{A_i}} \mu_E(y), \quad v_{\sum_{i=1}^{m} \underline{R_{A_i}}^{O}(E)}(x) = \overset{m}{\underset{i=1}{\wedge}} \sup_{y \in [x]_{A_i}} v_E(y);$$

$$\mu_{\overline{\sum_{i=1}^{m} R_{A_i}}^{O}(E)}(x) = \overset{m}{\underset{i=1}{\wedge}} \sup_{y \in [x]_{A_i}} \mu_E(y), \quad v_{\overline{\sum_{i=1}^{m} R_{A_i}}^{O}(E)}(x) = \overset{m}{\underset{i=1}{\vee}} \inf_{y \in [x]_{A_i}} v_E(y).$$

*where $R_{A_i}$ is an equivalence relation of $x$ in $A$, $[x]_{A_i}$ is the equivalence class of $R_{A_i}$, and $\vee$ is a disjunction operation.*

**Definition 12** ([22,25]). *Suppose $IS = <U, A, V, f>$ is an information system, $A = \{A_1, A_2, \cdots, A_m\}$. $\forall E \subseteq U$, $E$ are IFS. Then the lower and upper approximations of pessimistic MRIFS of $A_i$ can be described as follows:*

$$\sum_{i=1}^{m} R_{A_i}^{I}(E) = \{<x, \mu_{\underline{\sum_{i=1}^{m} R_{A_i}^{I}(E)}}(x), \nu_{\underline{\sum_{i=1}^{m} R_{A_i}^{I}(E))}}(x) > | x \in U\};$$

$$\overline{\sum_{i=1}^{m} R_{A_i}}^{I}(E) = \{<x, \mu_{\overline{\sum_{i=1}^{m} R_{A_i}}^{I}(E)}(x), \nu_{\overline{\sum_{i=1}^{m} R_{A_i}}^{I}(E)}(x) > | x \in U\}.$$

*where*

$$\mu_{\underline{\sum_{i=1}^{m} R_{A_i}^{I}(E)}}(x) = \bigwedge_{i=1}^{m} \inf_{y \in [x]_{A_i}} \mu_E(y), \quad \nu_{\underline{\sum_{i=1}^{m} R_{A_i}^{I}(E)}}(x) = \bigvee_{i=1}^{m} \sup_{y \in [x]_{A_i}} \nu_E(y);$$

$$\mu_{\overline{\sum_{i=1}^{m} R_{A_i}}^{I}(E)}(x) = \bigvee_{i=1}^{m} \sup_{y \in [x]_{A_i}} \mu_E(y), \quad \nu_{\overline{\sum_{i=1}^{m} R_{A_i}}^{I}(E)}(x) = \bigwedge_{i=1}^{m} \inf_{y \in [x]_{A_i}} \nu_E(y).$$

*where $[x]_{A_i}$ is the equivalence class of $x$ about the equivalence relation $R_{A_i}$, and $\wedge$ is a conjunction operation.*

**Definition 13.** *Suppose $IS = <U, A, V, f>$ is an information system, $A_i^O = \{A_1, A_2, \cdots, A_r\} \subseteq A$, $A = \{A_1, A_2, \cdots, A_m\}$. And $R_{A_i O}$ is an equivalence relation of $x$ with respect to the attribute reduction set $A_i^O$ under OMRS, $[x]_{A_i O}$ is the equivalence class of $R_{A_i O}$. Let $E$ be IFS of $U$ and they can be characterized by a pair of lower and upper approximations:*

$$\sum_{i=1}^{r} R_{A_i^O}^{O}(E) = \{<x, \mu_{\underline{\sum_{i=1}^{r} R_{A_i^O}^{O}(E)}}(x), \nu_{\underline{\sum_{i=1}^{r} R_{A_i^O}^{O}(E)}}(x) > | x \in U\};$$

$$\overline{\sum_{i=1}^{r} R_{A_i^O}}^{O}(E) = \{<x, \mu_{\overline{\sum_{i=1}^{r} R_{A_i^O}}^{O}(E)}(x), \nu_{\overline{\sum_{i=1}^{r} R_{A_i^O}}^{O}(E)}(x) > | x \in U\}.$$

*where*

$$\mu_{\underline{\sum_{i=1}^{r} R_{A_i^O}^{O}(E)}}(x) = \bigvee_{i=1}^{r} \inf_{y \in [x]_{A_i O}} \mu_E(y), \quad \nu_{\underline{\sum_{i=1}^{r} R_{A_i^O}^{O}(E)}}(x) = \bigwedge_{i=1}^{r} \sup_{y \in [x]_{A_i O}} \nu_E(y);$$

$$\mu_{\overline{\sum_{i=1}^{r} R_{A_i^O}}^{O}(E)}(x) = \bigwedge_{i=1}^{r} \sup_{y \in [x]_{A_i O}} \mu_E(y), \quad \nu_{\overline{\sum_{i=1}^{r} R_{A_i^O}}^{O}(E)}(x) = \bigvee_{i=1}^{r} \inf_{y \in [x]_{A_i O}} \nu_E(y).$$

*If $\sum_{i=1}^{r} R_{A_i^O}^{O}(E) \neq \overline{\sum_{i=1}^{r} R_{A_i^O}}^{O}(E)$, then $E$ can be called OOMRIFS.*

**Definition 14.** *Suppose $IS = <U, A, V, f>$ is an information system, $\forall E \subseteq U$, $E$ are IFS. $A_i^O = \{A_1, A_2, \cdots, A_r\} \subseteq A$, $A = \{A_1, A_2, \cdots, A_m\}$. where $A_i^O$ is an optimistic multi-granulation attribute reduction set. Then the lower and upper approximations of pessimistic MRIFS under optimistic multi-granulation environment can be defined as follows:*

$$\sum_{i=1}^{r} R_{A_i^O}^{I}(E) = \{<x, \mu_{\underline{\sum_{i=1}^{r} R_{A_i^O}^{I}(E)}}(x), \nu_{\underline{\sum_{i=1}^{r} R_{A_i^O}^{I}(E)}}(x) > | x \in U\};$$

$$\overline{\sum_{i=1}^{r} R_{A_i^O}}^{I}(E) = \{<x, \mu_{\overline{\sum_{i=1}^{r} R_{A_i^O}}^{I}(E)}(x), \nu_{\overline{\sum_{i=1}^{r} R_{A_i^O}}^{I}(E)}(x) > | x \in U\}.$$

where

$$\mu_{\underline{\sum\limits_{i=1}^{r} R_{A_i^O}}(E)}^{I}(x) = \bigwedge_{i=1}^{r}\inf_{y\in[x]_{A_i^O}}\mu_E(y), \nu_{\underline{\sum\limits_{i=1}^{r} R_{A_i^O}}(E)}^{I}(x) = \bigvee_{i=1}^{r}\sup_{y\in[x]_{A_i^O}}\nu_E(y);$$

$$\mu_{\overline{\sum\limits_{i=1}^{r} R_{A_i^O}}(E)}^{I}(x) = \bigvee_{i=1}^{r}\sup_{y\in[x]_{A_i^O}}\mu_E(y), \nu_{\overline{\sum\limits_{i=1}^{r} R_{A_i^O}}(E)}^{I}(x) = \bigwedge_{i=1}^{r}\inf_{y\in[x]_{A_i^O}}\nu_E(y).$$

*The pair* $(\underline{\sum\limits_{i=1}^{r} R_{A_i^O}}^{I}(E), \overline{\sum\limits_{i=1}^{r} R_{A_i^O}}^{I}(E))$ *are called OIMRIFS, if* $\underline{\sum\limits_{i=1}^{r} R_{A_i^O}}^{I}(E) \neq \overline{\sum\limits_{i=1}^{r} R_{A_i^O}}^{I}(E)$.

*According to Definitions 13 and 14, the following theorem can be obtained.*

**Theorem 4.** *Let* $IS = < U, A, V, f >$ *be an information system,* $A_i^O = \{A_1, A_2, \cdots, A_r\} \subseteq A$, $A = \{A_1, A_2, \cdots, A_m\}$, *and* $E_1, E_2$ *be IFS on U. Comparing with Definitions 13 and 14, the following proposition is obtained.*

(1) $\quad \underline{\sum\limits_{i=1}^{r} R_{A_i^O}}^{O}(E_1) = \bigcup\limits_{i=1}^{r}\underline{R_{A_i^O}}^{O}(E_1);$

(2) $\quad \overline{\sum\limits_{i=1}^{r} R_{A_i^O}}^{O}(E_1) = \bigcap\limits_{i=1}^{r}\overline{R_{A_i^O}}^{O}(E_1);$

(3) $\quad \underline{\sum\limits_{i=1}^{r} R_{A_i^O}}^{I}(E_1) = \bigcap\limits_{i=1}^{r}\underline{R_{A_i^O}}^{I}(E_1);$

(4) $\quad \overline{\sum\limits_{i=1}^{r} R_{A_i^O}}^{I}(E_1) = \bigcup\limits_{i=1}^{r}\overline{R_{A_i^O}}^{I}(E_1);$

(5) $\quad \underline{\sum\limits_{i=1}^{r} R_{A_i^O}}^{I}(E_1) \subseteq \underline{\sum\limits_{i=1}^{r} R_{A_i^O}}^{O}(E_1);$

(6) $\quad \overline{\sum\limits_{i=1}^{r} R_{A_i^O}}^{O}(E_1) \subseteq \overline{\sum\limits_{i=1}^{r} R_{A_i^O}}^{I}(E_1);$

(7) $\quad \underline{\sum\limits_{i=1}^{r} R_{A_i^O}}^{O}(E_1 \cap E_2) = \underline{\sum\limits_{i=1}^{r} R_{A_i^O}}^{O}(E_1) \cap \underline{\sum\limits_{i=1}^{r} R_{A_i^O}}^{O}(E_2), \underline{\sum\limits_{i=1}^{r} R_{A_i^O}}^{I}(E_1 \cap E_2) = \underline{\sum\limits_{i=1}^{r} R_{A_i^O}}^{I}(E_1) \cap \underline{\sum\limits_{i=1}^{r} R_{A_i^O}}^{I}(E_2);$

(8) $\quad \overline{\sum\limits_{i=1}^{r} R_{A_i^O}}^{O}(E_1 \cup E_2) = \overline{\sum\limits_{i=1}^{r} R_{A_i^O}}^{O}(E_1) \cup \overline{\sum\limits_{i=1}^{r} R_{A_i^O}}^{O}(E_2), \overline{\sum\limits_{i=1}^{r} R_{A_i^O}}^{I}(E_1 \cup E_2) = \overline{\sum\limits_{i=1}^{r} R_{A_i^O}}^{I}(E_1) \cup \overline{\sum\limits_{i=1}^{r} R_{A_i^O}}^{I}(E_2);$

(9) $\quad \underline{\sum\limits_{i=1}^{r} R_{A_i^O}}^{O}(E_1 \cup E_2) \supseteq \underline{\sum\limits_{i=1}^{r} R_{A_i^O}}^{O}(E_1) \cup \underline{\sum\limits_{i=1}^{r} R_{A_i^O}}^{O}(E_2), \underline{\sum\limits_{i=1}^{r} R_{A_i^O}}^{I}(E_1 \cup E_2) \supseteq \underline{\sum\limits_{i=1}^{r} R_{A_i^O}}^{I}(E_1) \cup \underline{\sum\limits_{i=1}^{r} R_{A_i^O}}^{I}(E_2);$

(10) $\quad \overline{\sum\limits_{i=1}^{r} R_{A_i^O}}^{O}(E_1 \cap E_2) \subseteq \overline{\sum\limits_{i=1}^{r} R_{A_i^O}}^{O}(E_1) \cap \overline{\sum\limits_{i=1}^{r} R_{A_i^O}}^{O}(E_2), \overline{\sum\limits_{i=1}^{r} R_{A_i^O}}^{I}(E_1 \cap E_2) \subseteq \overline{\sum\limits_{i=1}^{r} R_{A_i^O}}^{I}(E_1) \cap \overline{\sum\limits_{i=1}^{r} R_{A_i^O}}^{I}(E_2).$

**Proof.** It is easy to prove by the Definitions 13 and 14. $\square$

**Definition 15.** *Let* $IS = < U, A, V, f >$ *be an information system, and E be IFS on U.* $A_i^I = \{A_1, A_2, \cdots, A_r\} \subseteq A$, $A = \{A_1, A_2, \cdots, A_m\}$, *where* $A_i^I$ *is a pessimistic multi-granulation attribute reduction set. Then, the pessimistic optimistic lower and upper approximations of E with respect to equivalence relation* $R_{A_i^I}$ *are defined by the following formulas:*

$$\underline{\sum\limits_{i=1}^{r} R_{A_i^I}}^{O}(E) = \{< x, \mu_{\underline{\sum\limits_{i=1}^{r} R_{A_i^I}}(E)}^{o}(x), \nu_{\underline{\sum\limits_{i=1}^{r} R_{A_i^I}}(E)}^{o}(x) > | x \in U\};$$

$$\overline{\sum\limits_{i=1}^{r} R_{A_i^I}}^{O}(E) = \{< x, \mu_{\overline{\sum\limits_{i=1}^{r} R_{A_i^I}}(E)}^{o}(x), \nu_{\overline{\sum\limits_{i=1}^{r} R_{A_i^I}}(E)}^{o}(x) > | x \in U\}.$$

*where*

$$\mu_{\underline{\sum_{i=1}^{r} R_{A_i^I}}^O (E)}(x) = \overset{r}{\underset{i=1}{\vee}} \inf_{y\in[x]_{A_i^I}} \mu_E(y), \nu_{\underline{\sum_{i=1}^{r} R_{A_i^I}}^O (E)}(x) = \overset{r}{\underset{i=1}{\wedge}} \sup_{y\in[x]_{A_i^I}} \nu_E(y);$$

$$\mu_{\overline{\sum_{i=1}^{r} R_{A_i^I}}^O (E)}(x) = \overset{r}{\underset{i=1}{\wedge}} \sup_{y\in[x]_{A_i^I}} \mu_E(y), \nu_{\overline{\sum_{i=1}^{r} R_{A_i^I}}^O (E)}(x) = \overset{r}{\underset{i=1}{\vee}} \inf_{y\in[x]_{A_i^I}} \nu_E(y).$$

*If* $\underline{\sum_{i=1}^{r} R_{A_i^I}}^O (E) \neq \overline{\sum_{i=1}^{r} R_{A_i^I}}^O (E)$, *then E can be called IOMRIFS.*

**Definition 16.** *Let IS* $=< U, A, V, f >$ *be an information system, and E be IFS on U.* $A_i^I = \{A_1, A_2, \cdots, A_r\} \subseteq A$, $A = \{A_1, A_2, \cdots, A_m\}$, *where* $A_i^I$ *is a pessimistic multi-granulation attribute reduction set. Then, the pessimistic lower and upper approximations of E under IMRS are defined by the following formulas:*

$$\underline{\sum_{i=1}^{r} R_{A_i^I}}^I (E) = \{< x, \mu_{\underline{\sum_{i=1}^{r} R_{A_i^I}}^I (E)}(x), \nu_{\underline{\sum_{i=1}^{r} R_{A_i^I}}^I (E)}(x) > | x \in U\};$$

$$\overline{\sum_{i=1}^{r} R_{A_i^I}}^I (E) = \{< x, \mu_{\overline{\sum_{i=1}^{r} R_{A_i^I}}^I (E)}(x), \nu_{\overline{\sum_{i=1}^{r} R_{A_i^I}}^I (E)}(x) > | x \in U\}.$$

*where*

$$\mu_{\underline{\sum_{i=1}^{r} R_{A_i^I}}^I (E)}(x) = \overset{r}{\underset{i=1}{\wedge}} \inf_{y\in[x]_{A_i^I}} \mu_E(y), \nu_{\underline{\sum_{i=1}^{r} R_{A_i^I}}^I (E)}(x) = \overset{r}{\underset{i=1}{\vee}} \sup_{y\in[x]_{A_i^I}} \nu_E(y);$$

$$\mu_{\overline{\sum_{i=1}^{r} R_{A_i^I}}^I (E)}(x) = \overset{r}{\underset{i=1}{\vee}} \sup_{y\in[x]_{A_i^I}} \mu_E(y), \nu_{\overline{\sum_{i=1}^{r} R_{A_i^I}}^I (E)}(x) = \overset{r}{\underset{i=1}{\wedge}} \inf_{y\in[x]_{A_i^I}} \nu_E(y).$$

*where* $R_{A_i^I}$ *is an equivalence relation of x about the attribute reduction set* $A_i^I$ *under IMRS,* $[x]_{A_i^O}$ *is the equivalence class of* $R_{A_i^I}$.

*If* $\underline{\sum_{i=1}^{r} R_{A_i^I}}^I (E) \neq \overline{\sum_{i=1}^{r} R_{A_i^I}}^I (E)$, *then the pair* $(\underline{\sum_{i=1}^{r} R_{A_i^I}}^I (E), \overline{\sum_{i=1}^{r} R_{A_i^I}}^I (E))$ *is said to be IIMRIFS.*

According to Definitions 15 and 16, the following theorem can be captured.

**Theorem 5.** *Let IS* $=< U, A, V, f >$ *be an information system,* $A_i^I = \{A_1, A_2, \cdots, A_r\} \subseteq A$, $A = \{A_1, A_2, \cdots, A_m\}$, *and* $E_1, E_2$ *be IFS on U. Then IOMRIFS and IIOMRIFS models have the following properties:*

(1) $\underline{\sum_{i=1}^{r} R_{A_i^I}}^O (E_1) = \overset{r}{\underset{i=1}{\cup}} \underline{R_{A_i^I}}^O (E_1);$

(2) $\overline{\sum_{i=1}^{r} R_{A_i^I}}^O (E_1) = \overset{r}{\underset{i=1}{\cap}} \overline{R_{A_i^I}}^O (E_1);$

(3) $\underline{\sum_{i=1}^{r} R_{A_i^I}}^I (E_1) = \overset{r}{\underset{i=1}{\cup}} \underline{R_{A_i^I}}^I (E_1);$

(4) $\overline{\sum_{i=1}^{r} R_{A_i^I}}^I (E_1) = \overset{r}{\underset{i=1}{\cup}} \underline{R_{A_i^I}}^I (E_1);$

(5) $\underline{\sum_{i=1}^{r} R_{A_i^I}}^I (E_1) \subseteq \underline{\sum_{i=1}^{r} R_{A_i^I}}^O (E_1);$

(6) $\overline{\sum_{i=1}^{r} R_{A_i^I}}^O (E_1) \subseteq \overline{\sum_{i=1}^{r} R_{A_i^I}}^I (E_1).$

(7) $\underline{\sum_{i=1}^{r} R_{A_i^I}}^O (E_1 \cap E_2) = \underline{\sum_{i=1}^{r} R_{A_i^I}}^O (E_1) \cap \underline{\sum_{i=1}^{r} R_{A_i^I}}^O (E_2), \underline{\sum_{i=1}^{r} R_{A_i^I}}^I (E_1 \cap E_2) = \underline{\sum_{i=1}^{r} R_{A_i^I}}^I (E_1) \cap \underline{\sum_{i=1}^{r} R_{A_i^I}}^I (E_2);$

(8) $\overline{\sum_{i=1}^{r} R_{A_i^I}}^O (E_1 \cup E_2) = \overline{\sum_{i=1}^{r} R_{A_i^I}}^O (E_1) \cup \overline{\sum_{i=1}^{r} R_{A_i^I}}^O (E_2), \overline{\sum_{i=1}^{r} R_{A_i^I}}^I (E_1 \cup E_2) = \overline{\sum_{i=1}^{r} R_{A_i^I}}^I (E_1) \cup \overline{\sum_{i=1}^{r} R_{A_i^I}}^I (E_2);$

(9) $\quad \sum\limits_{i=1}^{r} R_{A_i^I}^{O}(E_1 \cup E_2) \supseteq \sum\limits_{i=1}^{r} R_{A_i^I}^{O}(E_1) \cup \sum\limits_{i=1}^{r} R_{A_i^I}^{O}(E_2), \sum\limits_{i=1}^{r} R_{A_i^I}^{I}(E_1 \cup E_2) \supseteq \sum\limits_{i=1}^{r} R_{A_i^I}^{I}(E_1) \cup \sum\limits_{i=1}^{r} R_{A_i^I}^{I}(E_2);$

(10) $\quad \overline{\sum\limits_{i=1}^{r} R_{A_i^I}}^{O}(E_1 \cap E_2) \subseteq \overline{\sum\limits_{i=1}^{r} R_{A_i^I}}^{O}(E_1) \cap \overline{\sum\limits_{i=1}^{r} R_{A_i^I}}^{O}(E_2), \overline{\sum\limits_{i=1}^{r} R_{A_i^I}}^{I}(E_1 \cap E_2) \subseteq \overline{\sum\limits_{i=1}^{r} R_{A_i^I}}^{I}(E_1) \cap \overline{\sum\limits_{i=1}^{r} R_{A_i^I}}^{I}(E_2).$

**Proof.** It can be derived directly from Definitions 15 and 16. $\quad\square$

The characteristics of the proposed four models are further verified by Example 2 below.

**Example 2.** (Continued with Example 1). *From Example 1, we know that these 5 sites are evaluated by 10 investment schemes respectively. Suppose they have the following IFS with respect to 10 investment schemes*

$$E = \left\{ \frac{[0.25,0.43]}{x_1}, \frac{[0.51,0.28]}{x_2}, \frac{[0.54,0.38]}{x_3}, \frac{[0.37,0.59]}{x_4}, \frac{[0.49,0.35]}{x_5}, \frac{[0.92,0.04]}{x_6}, \frac{[0.09,0.86]}{x_7}, \frac{[0.15,0.46]}{x_8}, \right.$$
$$\left. \frac{[0.72,0.12]}{x_9}, \frac{[0.67,0.23]}{x_{10}} \right\}.$$

(1) *In OOMRIFS, the lower and upper approximations of OOMRIFS can be calculated as follows:*

$$\underline{\sum\limits_{i=1}^{r} R_{A_i^O}}^{O}(E) = \left\{ \frac{[0.25,0.59]}{x_1}, \frac{[0.49,0.38]}{x_2}, \frac{[0.49,0.38]}{x_3}, \frac{[0.25,0.59]}{x_4}, \frac{[0.49,0.38]}{x_5}, \frac{[0.25,0.46]}{x_6}, \frac{[0.09,0.86]}{x_7}, \right.$$
$$\left. \frac{[0.15,0.46]}{x_8}, \frac{[0.15,0.46]}{x_9}, \frac{[0.67,0.23]}{x_{10}} \right\},$$

$$\overline{\sum\limits_{i=1}^{r} R_{A_i^O}}^{O}(E) = \left\{ \frac{[0.51,0.28]}{x_1}, \frac{[0.51,0.28]}{x_2}, \frac{[0.54,0.35]}{x_3}, \frac{[0.51,0.28]}{x_4}, \frac{[0.54,0.35]}{x_5}, \frac{[0.92,0.04]}{x_6}, \frac{[0.54,0.35]}{x_7}, \right.$$
$$\left. \frac{[0.15,0.46]}{x_8}, \frac{[0.72,0.12]}{x_9}, \frac{[0.67,0.23]}{x_{10}} \right\}.$$

(2) *Similarly, in OIMRIFS, we have:*

$$\underline{\sum\limits_{i=1}^{r} R_{A_i^O}}^{I}(E) = \left\{ \frac{[0.25,0.59]}{x_1}, \frac{[0.25,0.59]}{x_2}, \frac{[0.09,0.86]}{x_3}, \frac{[0.25,0.59]}{x_4}, \frac{[0.09,0.86]}{x_5}, \frac{[0.15,0.59]}{x_6}, \frac{[0.09,0.86]}{x_7}, \right.$$
$$\left. \frac{[0.15,0.46]}{x_8}, \frac{[0.09,0.86]}{x_9}, \frac{[0.09,0.86]}{x_{10}} \right\},$$

$$\overline{\sum\limits_{i=1}^{r} R_{A_i^O}}^{I}(E) = \left\{ \frac{[0.92,0.04]}{x_1}, \frac{[0.54,0.28]}{x_2}, \frac{[0.54,0.28]}{x_3}, \frac{[0.92,0.04]}{x_4}, \frac{[0.54,0.28]}{x_5}, \frac{[0.92,0.04]}{x_6}, \frac{[0.72,0.12]}{x_7}, \right.$$
$$\left. \frac{[0.92,0.04]}{x_8}, \frac{[0.92,0.04]}{x_9}, \frac{[0.72,0.12]}{x_{10}} \right\}.$$

*From the above results, Figure 1 can be drawn as follows:*

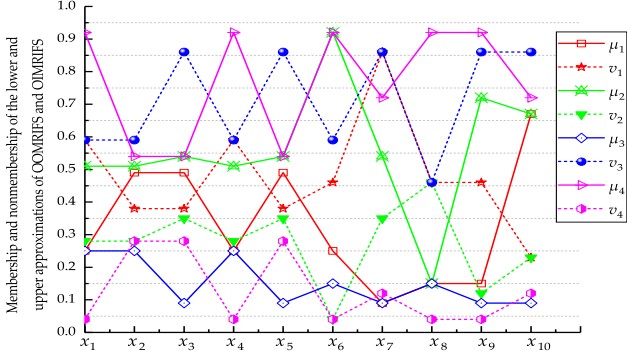

**Figure 1.** The lower and upper approximations of OOMRIFS and OIMRIFS.

*Note that*
$\mu_1 = \underline{\mu^{OO}}(x_j)$ *and* $\nu_1 = \underline{\nu^{OO}}(x_j)$ *represent the lower approximation of OOMRIFS;*
$\mu_2 = \overline{\mu^{OO}}(x_j)$ *and* $\nu_2 = \overline{\nu^{OO}}(x_j)$ *represent the upper approximation of OOMRIFS;*

$\mu_3 = \underline{\mu^{OI}}(x_j)$ and $\nu_3 = \underline{\nu^{OI}}(x_j)$ *represent the lower approximation of OIMRIFS;*

$\mu_4 = \overline{\mu^{OI}}(x_j)$ and $\nu_4 = \overline{\nu^{OI}}(x_j)$ *represent the upper approximation of OIMRIFS.*

*Regarding Figure 1, we can get,*

$$\overline{\mu^{OI}}(x_j) \geq \overline{\mu^{OO}}(x_j) \geq \underline{\mu^{OO}}(x_j) \geq \underline{\mu^{OI}}(x_j); \ \underline{\nu^{OI}}(x_j) \geq \underline{\nu^{OO}}(x_j) \geq \overline{\nu^{OO}}(x_j) \geq \overline{\nu^{OI}}(x_j).$$

*As shown in Figure 1, the rules of Theorem 4 are satisfied. By constructing the OOMRIFS and OIMRIFS models, we can reduce the subjective scoring errors of experts under intuitionistic fuzzy conditions.*

*(3) Similar to (1), in IOMRIFS, we have:*

$$\underline{\sum_{i=1}^{r} R_{A_i^I}}^{O}(E) = \left\{ \frac{[0.25,0.43]}{x_1}, \frac{[0.25,0.43]}{x_2}, \frac{[0.25,0.43]}{x_3}, \frac{[0.37,0.59]}{x_4}, \frac{[0.25,0.43]}{x_5}, \frac{[0.25,0.46]}{x_6}, \frac{[0.09,0.86]}{x_7}, \right.$$
$$\left. \frac{[0.15,0.46]}{x_8}, \frac{[0.67,0.23]}{x_9}, \frac{[0.67,0.23]}{x_{10}} \right\},$$

$$\overline{\sum_{i=1}^{r} R_{A_i^I}}^{O}(E) = \left\{ \frac{[0.51,0.28]}{x_1}, \frac{[0.51,0.28]}{x_2}, \frac{[0.54,0.35]}{x_3}, \frac{[0.37,0.59]}{x_4}, \frac{[0.49,0.35]}{x_5}, \frac{[0.92,0.04]}{x_6}, \frac{[0.51,0.35]}{x_7}, \right.$$
$$\left. \frac{[0.49,0.35]}{x_8}, \frac{[0.72,0.12]}{x_9}, \frac{[0.67,0.23]}{x_{10}} \right\}.$$

*(4) The same as (1), in IIMRIFS, we can get:*

$$\underline{\sum_{i=1}^{r} R_{A_i^I}}^{I}(E) = \left\{ \frac{[0.25,0.59]}{x_1}, \frac{[0.09,0.86]}{x_2}, \frac{[0.09,0.86]}{x_3}, \frac{[0.25,0.59]}{x_4}, \frac{[0.09,0.86]}{x_5}, \frac{[0.09,0.86]}{x_6}, \frac{[0.09,0.86]}{x_7}, \right.$$
$$\left. \frac{[0.09,0.86]}{x_8}, \frac{[0.15,0.46]}{x_9}, \frac{[0.67,0.23]}{x_{10}} \right\},$$

$$\overline{\sum_{i=1}^{r} R_{A_i^I}}^{I}(E) = \left\{ \frac{[0.92,0.04]}{x_1}, \frac{[0.54,0.28]}{x_2}, \frac{[0.92,0.04]}{x_3}, \frac{[0.92,0.04]}{x_4}, \frac{[0.54,0.28]}{x_5}, \frac{[0.92,0.04]}{x_6}, \frac{[0.92,0.04]}{x_7}, \right.$$
$$\left. \frac{[0.92,0.04]}{x_8}, \frac{[0.92,0.04]}{x_9}, \frac{[0.72,0.12]}{x_{10}} \right\}.$$

*From (3) and (4), we can obtain Figure 2 as shown:*

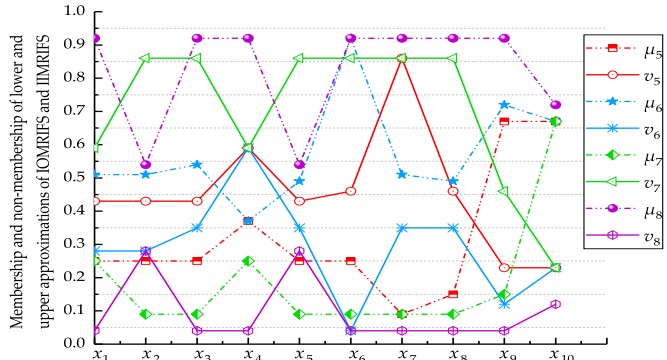

**Figure 2.** The lower and upper approximations of IOMRIFS and IIMRIFS.

*Note that*

$\mu_5 = \underline{\mu^{IO}}(x_j)$ and $\nu_5 = \underline{\nu^{IO}}(x_j)$ *represent the lower approximation of IOMRIFS;*

$\mu_6 = \overline{\mu^{IO}}(x_j)$ and $\nu_6 = \overline{\nu^{IO}}(x_j)$ *represent the upper approximation of IOMRIFS;*

$\mu_7 = \underline{\mu^{II}}(x_j)$ and $\nu_7 = \underline{\nu^{II}}(x_j)$ *represent the lower approximation of IIMRIFS;*

$\mu_8 = \overline{\mu^{II}}(x_j)$ and $\nu_8 = \overline{\nu^{II}}(x_j)$ *represent the upper approximation of IIMRIFS.*

*For Figure 2, we can get,*

$$\overline{\mu^{II}}(x_j) \geq \overline{\mu^{IO}}(x_j) \geq \underline{\mu^{IO}}(x_j) \geq \underline{\mu^{II}}(x_j); \ \underline{\nu^{II}}(x_j) \geq \underline{\nu^{IO}}(x_j) \geq \overline{\nu^{IO}}(x_j) \geq \overline{\nu^{II}}(x_j).$$

*As shown in Figure 2, the rules of Theorem 5 are satisfied.*

Through the Example 2, we can obtain four relatively more objective MRIFS models, which are beneficial to reduce subjective errors.

## 5. Three-Way Decisions Models Based on MRIFS and Optimal Granularity Selection

In order to obtain the optimal granularity selection results in the case of optimistic and pessimistic multi-granulation sets, it is necessary to further distinguish the importance degree of each granularity in the reduction sets. We respectively combine the four MRIFS models mentioned above with three-way decisions theory to get four new three-way decisions models. By extracting the rules, the redundant objects in the reduction sets are removed, and the decision error is further reduced. Then the optimal granularity selection results in two cases are obtained respectively by constructing the comprehensive score function and comprehensive accuracy function measurement formulas of each granularity of the reduction sets.

### 5.1. Three-Way Decisions Model Based on OOMRIFS

Suppose $A_i^O$ is the reduction set under OMRS. According to reference [46], the expected loss function $R^{OO}(\omega_* | [x]_{A_i^O})(* = P, B, N)$ of object x can be obtained:

$$R^{OO}(\omega_P | [x]_{A_i^O}) = \lambda_{PP} \cdot \mu^{OO}(x) + \lambda_{PN} \cdot \nu^{OO}(x) + \lambda_{PB} \cdot \pi^{OO}(x);$$
$$R^{OO}(\omega_N | [x]_{A_i^O}) = \lambda_{NP} \cdot \mu^{OO}(x) + \lambda_{NN} \cdot \nu^{OO}(x) + \lambda_{NB} \cdot \pi^{OO}(x);$$
$$R^{OO}(\omega_B | [x]_{A_i^O}) = \lambda_{BP} \cdot \mu^{OO}(x) + \lambda_{BN} \cdot \nu^{OO}(x) + \lambda_{BB} \cdot \pi^{OO}(x).$$

where

$$\mu^{OO}(x) = \mu_{\sum\limits_{i=1}^r R_{A_i^O}}{}^O(E)(x) = \bigvee\limits_{i=1}^r \inf\limits_{y \in [x]_{A_i^O}} \mu_E(y), \nu^{OO}(x) = \nu_{\sum\limits_{i=1}^r R_{A_i^O}}{}^O(E)(x) = \bigwedge\limits_{i=1}^r \sup\limits_{y \in [x]_{A_i^O}} \nu_E(y), \pi^{OO}(x) = 1 - \mu_{\sum\limits_{i=1}^r R_{A_i^O}}{}^O(E)(x) - \nu_{\sum\limits_{i=1}^r R_{A_i^O}}{}^O(E)(x);$$

or

$$\mu^{OO}(x) = \mu_{\overline{\sum\limits_{i=1}^r R_{A_i^O}}}{}^O(E)(x) = \bigwedge\limits_{i=1}^r \sup\limits_{y \in [x]_{A_i^O}} \mu_E(y), \nu^{OO}(x) = \nu_{\overline{\sum\limits_{i=1}^r R_{A_i^O}}}{}^O(E)(x) = \bigvee\limits_{i=1}^r \inf\limits_{y \in [x]_{A_i^O}} \nu_E(y), \pi^{OO}(x) = 1 - \mu_{\overline{\sum\limits_{i=1}^r R_{A_i^O}}}{}^O(E)(x) - \nu_{\overline{\sum\limits_{i=1}^r R_{A_i^O}}}{}^O(E)(x).$$

The minimum-risk decision rules derived from the Bayesian decision process are as follows:

(P′): If $R'(\omega_P | [x]_{A_i^O}) \leq R'(\omega_B | [x]_{A_i^O})$ and $R'(\omega_P | [x]_{A_i^O}) \leq R'(\omega_N | [x]_{A_i^O})$, then $x \in POS(X)$;

(N′): If $R'(\omega_N | [x]_{A_i^O}) \leq R'(\omega_P | [x]_{A_i^O})$ and $R'(\omega_N | [x]_{A_i^O}) \leq R'(\omega_B | [x]_{A_i^O})$, then $x \in NEG(X)$;

(B′): If $R'(\omega_B | [x]_{A_i^O}) \leq R'(\omega_N | [x]_{A_i^O})$ and $R'(\omega_B | [x]_{A_i^O}) \leq R'(\omega_P | [x]_{A_i^O})$, then $x \in BND(X)$.

Thus, the decision rules (P′)-(B′) can be re-expressed concisely as:

(P′) rule satisfies:

$$(\mu^{OO}(x) \leq (1 - \pi^{OO}(x)) \cdot \frac{\lambda_{NN} - \lambda_{PN}}{(\lambda_{PP} - \lambda_{NP}) + (\lambda_{PN} - \lambda_{NN})}) \wedge (\mu^{OO}(x) \leq (1 - \pi^{OO}(x)) \cdot \frac{\lambda_{BN} - \lambda_{PN}}{(\lambda_{PP} - \lambda_{BP}) + (\lambda_{PN} - \lambda_{BN})});$$

(N′) rule satisfies:

$$(\mu^{OO}(x) < (1 - \pi^{OO}(x)) \cdot \frac{\lambda_{PN} - \lambda_{NN}}{(\lambda_{NP} - \lambda_{PP}) + (\lambda_{PN} - \lambda_{NN})}) \wedge (\mu^{OO}(x) < (1 - \pi^{OO}(x)) \cdot \frac{\lambda_{BN} - \lambda_{NN}}{(\lambda_{NP} - \lambda_{BP}) + (\lambda_{BN} - \lambda_{NN})});$$

(B′) rule satisfies:

$$(\mu^{OO}(x) > (1 - \pi^{OO}(x)) \cdot \frac{\lambda_{BN} - \lambda_{PN}}{(\lambda_{PN} - \lambda_{BN}) + (\lambda_{BP} - \lambda_{PP})}) \wedge (\mu^{OO}(x) \geq (1 - \pi^{OO}(x)) \cdot \frac{\lambda_{BN} - \lambda_{NN}}{(\lambda_{BN} - \lambda_{NN}) + (\lambda_{NP} - \lambda_{BP})}).$$

Therefore, the three-way decisions rules based on OOMRIFS are as follows:

(P1): If $\mu^{OO}(x) \geq (1 - \pi^{OO}(x)) \cdot \alpha$, then $x \in POS(X)$;

(N1): If $\mu^{OO}(x) \leq (1 - \pi^{OO}(x)) \cdot \beta$, then $x \in NEG(X)$;

(B1): If $(1 - \pi^{OO}(x)) \cdot \beta \leq \mu^{OO}(x)$ and $\mu^{OO}(x) \leq (1 - \pi^{OO}(x)) \cdot \alpha$, then $x \in BND(X)$.

### 5.2. Three-Way Decisions Model Based on OIMRIFS

Suppose $A_i^O$ is the reduction set under OMRS. According to reference [46], the expected loss functions $R^{OO}(\omega_*|[x]_{A_i^O})(* = P, B, N)$ of an object $x$ are presented as follows:

$$R^{OI}(\omega_P|[x]_{A_i^O}) = \lambda_{PP} \cdot \mu^{OI}(x) + \lambda_{PN} \cdot \nu^{OI}(x) + \lambda_{PB} \cdot \pi^{OI}(x);$$
$$R^{OI}(\omega_N|[x]_{A_i^O}) = \lambda_{NP} \cdot \mu^{OI}(x) + \lambda_{NN} \cdot \nu^{OI}(x) + \lambda_{NB} \cdot \pi^{OI}(x);$$
$$R^{OI}(\omega_B|[x]_{A_i^O}) = \lambda_{BP} \cdot \mu^{OI}(x) + \lambda_{BN} \cdot \nu^{OI}(x) + \lambda_{BB} \cdot \pi^{OI}(x).$$

where

$$\mu^{OI}(x) = \mu_{\sum_{i=1}^{r} R_{A_i^O}^I (E)}(x) = \bigwedge_{i=1}^{r} \inf_{y \in [x]_{A_i^O}} \mu_E(y), \nu^{OI}(x) = \nu_{\sum_{i=1}^{r} R_{A_i^O}^I (E)}(x) = \bigvee_{i=1}^{r} \sup_{y \in [x]_{A_i^O}} \nu_E(y), \pi^{OI}(x) = 1 - \mu_{\sum_{i=1}^{r} R_{A_i^O}^I (E)}(x) - \nu_{\sum_{i=1}^{r} R_{A_i^O}^I (E)}(x);$$

or

$$\mu^{OI}(x) = \mu_{\overline{\sum_{i=1}^{r} R_{A_i^O}^I} (E)}(x) = \bigvee_{i=1}^{r} \sup_{y \in [x]_{A_i^O}} \mu_E(y), \nu^{OI}(x) = \nu_{\overline{\sum_{i=1}^{r} R_{A_i^O}^I} (E)}(x) = \bigwedge_{i=1}^{r} \inf_{y \in [x]_{A_i^O}} \nu_E(y), \pi^{OI}(x) = 1 - \mu_{\overline{\sum_{i=1}^{r} R_{A_i^O}^I} (E)}(x) - \nu_{\overline{\sum_{i=1}^{r} R_{A_i^O}^I} (E)}(x).$$

Therefore, the three-way decisions rules based on OIMRIFS are as follows:
(P2): If $\mu^{OI}(x) \geq (1 - \pi^{OI}(x)) \cdot \alpha$, then $x \in POS(X)$;
(N2): If $\mu^{OI}(x) \leq (1 - \pi^{OI}(x)) \cdot \beta$, then $x \in NEG(X)$;
(B2): If $(1 - \pi^{OI}(x)) \cdot \beta \leq \mu^{OI}(x)$ and $\mu^{OI}(x) \leq (1 - \pi^{OI}(x)) \cdot \alpha$, then $x \in BND(X)$.

### 5.3. Three-Way Decisions Model Based on IOMRIFS

Suppose $A_i^I$ is the reduction set under IMRS. According to reference [46], the expected loss functions $R^{IO}(\omega_*|[x]_{A_i^I})(* = P, B, N)$ of an object $x$ are as follows:

$$R^{IO}(\omega_P|[x]_{A_i^I}) = \lambda_{PP} \cdot \mu^{IO}(x) + \lambda_{PN} \cdot \nu^{IO}(x) + \lambda_{PB} \cdot \pi^{IO}(x);$$
$$R^{IO}(\omega_N|[x]_{A_i^I}) = \lambda_{NP} \cdot \mu^{IO}(x) + \lambda_{NN} \cdot \nu^{IO}(x) + \lambda_{NB} \cdot \pi^{IO}(x);$$
$$R^{IO}(\omega_B|[x]_{A_i^I}) = \lambda_{BP} \cdot \mu^{IO}(x) + \lambda_{BN} \cdot \nu^{IO}(x) + \lambda_{BB} \cdot \pi^{IO}(x).$$

where

$$\mu^{IO}(x) = \mu_{\sum_{i=1}^{r} R_{A_i^I}^O (E)}(x) = \bigvee_{i=1}^{r} \inf_{y \in [x]_{A_i^I}} \mu_E(y), \nu^{IO}(x) = \nu_{\sum_{i=1}^{r} R_{A_i^I}^O (E)}(x) = \bigwedge_{i=1}^{r} \sup_{y \in [x]_{A_i^I}} \nu_E(y), \pi^{IO}(x) = 1 - \mu_{\sum_{i=1}^{r} R_{A_i^I}^O (E)}(x) - \nu_{\sum_{i=1}^{r} R_{A_i^I}^O (E)}(x);$$

or

$$\mu^{IO}(x) = \mu_{\overline{\sum_{i=1}^{r} R_{A_i^I}^O} (E)}(x) = \bigwedge_{i=1}^{r} \sup_{y \in [x]_{A_i^I}} \mu_E(y), \nu^{IO}(x) = \nu_{\overline{\sum_{i=1}^{r} R_{A_i^I}^O} (E)}(x) = \bigvee_{i=1}^{r} \inf_{y \in [x]_{A_i^I}} \nu_E(y), \pi^{IO}(x) = 1 - \mu_{\overline{\sum_{i=1}^{r} R_{A_i^I}^O} (E)}(x) - \nu_{\overline{\sum_{i=1}^{r} R_{A_i^I}^O} (E)}(x).$$

Therefore, the three-way decisions rules based on IOMRIFS are as follows:
(P3): If $\mu^{IO}(x) \geq (1 - \pi^{IO}(x)) \cdot \alpha$, then $x \in POS(X)$;
(N3): If $\mu^{IO}(x) \leq (1 - \pi^{IO}(x)) \cdot \beta$, then $x \in NEG(X)$;
(B3): If $(1 - \pi^{IO}(x)) \cdot \beta \leq \mu^{IO}(x)$ and $\mu^{IO}(x) \leq (1 - \pi^{IO}(x)) \cdot \alpha$, then $x \in BND(X)$.

### 5.4. Three-Way Decisions Model Based on IIMRIFS

Suppose $A_i^I$ is the reduction set under IMRS. Like Section 5.1, the expected loss functions $R^{II}(\omega_*|[x]_{A_i^I})(* = P, B, N)$ of an object $x$ are as follows:

$$R^{II}(\omega_P|[x]_{A_i^I}) = \lambda_{PP} \cdot \mu^{II}(x) + \lambda_{PN} \cdot \nu^{II}(x) + \lambda_{PB} \cdot \pi^{II}(x);$$

$$R^{II}(\omega_N|[x]_{A_i^I}) = \lambda_{NP} \cdot \mu^{II}(x) + \lambda_{NN} \cdot \nu^{II}(x) + \lambda_{NB} \cdot \pi^{II}(x);$$

$$R^{II}(\omega_B|[x]_{A_i^I}) = \lambda_{BP} \cdot \mu^{II}(x) + \lambda_{BN} \cdot \nu^{II}(x) + \lambda_{BB} \cdot \pi^{II}(x).$$

where

$$\mu^{II}(x) = \mu_{\underset{i=1}{\overset{r}{\Sigma}} R_{A_i^I}(E)}(x) = \overset{r}{\underset{i=1}{\wedge}} \underset{y \in [x]_{A_i^I}}{\inf} \mu_E(y), \quad \nu^{II}(x) = \nu_{\underset{i=1}{\overset{r}{\Sigma}} R_{A_i^I}(E)}(x) = \overset{r}{\underset{i=1}{\vee}} \underset{y \in [x]_{A_i^I}}{\sup} \nu_E(y), \quad \pi^{II}(x) = 1 - \mu_{\underset{i=1}{\overset{r}{\Sigma}} R_{A_i^I}(E)}(x) - \nu_{\underset{i=1}{\overset{r}{\Sigma}} R_{A_i^I}(E)}(x);$$

or

$$\mu^{II}(x) = \mu_{\underset{i=1}{\overset{r}{\Sigma}} R_{A_i^I}(E)}(x) = \overset{r}{\underset{i=1}{\vee}} \underset{y \in [x]_{A_i^I}}{\sup} \mu_E(y), \quad \nu^{II}(x) = \nu_{\underset{i=1}{\overset{r}{\Sigma}} R_{A_i^I}(E)}(x) = \overset{r}{\underset{i=1}{\wedge}} \underset{y \in [x]_{A_i^I}}{\inf} \nu_E(y), \quad \pi^{II}(x) = 1 - \mu_{\underset{i=1}{\overset{r}{\Sigma}} R_{A_i^I}(E)}(x) - \nu_{\underset{i=1}{\overset{r}{\Sigma}} R_{A_i^I}(E)}(x).$$

Therefore, the three-way decisions rules based on IIMRIFS are captured as follows:

(P4): If $\mu^{II}(x) \geq (1 - \pi^{II}(x)) \cdot \alpha$, then $x \in POS(X)$;

(N4): If $\mu^{II}(x) \leq (1 - \pi^{II}(x)) \cdot \beta$, then $x \in NEG(X)$;

(B4): If $(1 - \pi^{II}(x)) \cdot \beta \leq \mu^{II}(x)$ and $\mu^{II}(x) \leq (1 - \pi^{II}(x)) \cdot \alpha$, then $x \in BND(X)$.

By constructing the above three decision models, the redundant objects in the reduction sets can be removed, which is beneficial to the optimal granular selection.

*5.5. Comprehensive Measuring Methods of Granularity*

**Definition 17** ([40]). *Let an intuitionistic fuzzy number $\widetilde{E}(f_1) = (\mu_{\widetilde{E}}(f_1), \nu_{\widetilde{E}}(f_1))$, $f_1 \in U$, then the score function of $\widetilde{E}(f_1)$ is calculated as:*

$$S(\widetilde{E}(f_1)) = \mu_{\widetilde{E}}(f_1) - \nu_{\widetilde{E}}(f_1).$$

*The accuracy function of $\widetilde{E}(f_1)$ is defined as:*

$$H(\widetilde{E}(f_1)) = \mu_{\widetilde{E}}(f_1) + \nu_{\widetilde{E}}(f_1).$$

*where $-1 \leq S(\widetilde{E}(f_1)) \leq 1$ and $0 \leq H(\widetilde{E}(f_1)) \leq 1$.*

**Definition 18.** *Let $DIS = (U, C \cup D)$ be a decision information system, $A = \{A_1, A_2, \cdots, A_m\}$ are m sub-attributes of C. Suppose E are IFS on the universe $U = \{x_1, x_2, \cdots, x_n\}$, defined by $\mu_{A_i}(x_j)$ and $\nu_{A_i}(x_j)$, where $\mu_{A_i}(x_j)$ and $\nu_{A_i}(x_j)$ are their membership and non-membership functions respectively. $|[x_j]_{A_i}|$ is the number of equivalence classes of $x_j$ on granularity $A_i$, $U/D = \{X_1, X_2, \cdots, X_s\}$ is the partition induced by the decision attributes D. Then, the comprehensive score function of granularity $A_i$ is captured as:*

$$CSF_{A_i}(E) = \frac{1}{s} \times \sum_{j=1, n \in [x_j]_{A_i}}^{n} \frac{|\mu_{A_i}(x_j) - \nu_{A_i}(x_j)|}{|[x_j]_{A_i}|}.$$

*The comprehensive accuracy function of granularity $A_i$ is captured as:*

$$CAF_{A_i}(E) = \frac{1}{s} \times \sum_{j=1, n \in [x_j]_{A_i}}^{n} \frac{|\mu_{A_i}(x_j) + \nu_{A_i}(x_j)|}{|[x_j]_{A_i}|}.$$

*where $-1 \leq CSF_{A_i}(E) \leq 1$ and $0 \leq CAF_{A_i}(E) \leq 1$.*

With respect to Definition 19, according to references [27,39], we can deduce the following rules.

**Definition 19.** *Let two granularities $A_1$, $A_2$, then we have:*

(1) *If $CSF_{A_1}(E) > CSF_{A_2}(E)$, then $A_2$ is smaller than $A_1$, expressed as $A_1 > A_2$;*

(2) *If $CSF_{A_1}(E) < CSF_{A_2}(E)$, then $A_1$ is smaller than $A_2$, expressed as $A_1 < A_2$;*

(3) *If $CSF_{A_1}(E) = CSF_{A_2}(E)$, then*

    (i) *If $CSF_{A_1}(E) = CSF_{A_2}(E)$, then $A_2$ is equal to $A_1$, expressed as $A_1 = A_2$;*

    (ii) *If $CSF_{A_1}(E) > CSF_{A_2}(E)$, then $A_2$ is smaller than $A_1$, expressed as $A_1 > A_2$;*

    (iii) *If $CSF_{A_1}(E) < CSF_{A_2}(E)$, then $A_1$ is smaller than $A_2$, expressed as $A_1 < A_2$.*

*5.6. Optimal Granularity Selection Algorithm to Derive Three-Way Decisions from MRIFS*

Suppose the reduction sets of optimistic and IMRS are $A_i^O$ and $A_i^I$ respectively. In this section, we take the reduction set under OMRS as an example to make the result $A_i^{O\prime}$ of optimal granularity selection.

---

**Algorithm 2.** Optimal granularity selection algorithm to derive three-way decisions from MRIFS

---

**Input:** $DIS = (U, C \cup D, V, f)$, $A = \{A_1, A_2, \cdots, A_m\}$ be m sub-attributes of condition attributes $C$, $\forall A_i \in A'$, $U/D = \{X_1, X_2, \cdots, X_s\}$, IFS $E$;
**Output:** Optimal granularity selection result $A_i^{O\prime}$.
1: compute via Algorithm 1;
2: **if** $|A_i^O| > 1$
3:　　**for** $\forall A_i \in A_i^O$
4:　　　　compute $\mu_{\sum_{i=1}^{r} R_{A_i^O}^{\Delta}(E)}(x_j)$, $\nu_{\sum_{i=1}^{r} R_{A_i^O}^{\Delta}(E)}(x_j)$, $\mu_{\overline{\sum_{i=1}^{r} R_{A_i^O}^{\Delta}}(E)}(x_j)$ and $\nu_{\overline{\sum_{i=1}^{r} R_{A_i^O}^{\Delta}}(E)}(x_j)$;
5:　　　　according (P1)-(B1) and (P2)-(B2), compute $POS(\underline{X^{O\Delta}})$, $NEG(\underline{X^{O\Delta}})$, $BND(\underline{X^{O\Delta}})$, $POS(\overline{X^{O\Delta}})$, $NEG(\overline{X^{O\Delta}})$, $BND(\overline{X^{O\Delta}})$;
6:　　　　**if** $NEG(X^{O\Delta}) \neq U$ or $NEG(\overline{X^{O\Delta}}) \neq U$
7:　　　　　　compute $U/\underline{A_i^{O\Delta}}$, $CSF_{\underline{A_i^{O\Delta}}}(E)$, $CAF_{\underline{A_i^{O\Delta}}}(E)$ or $(U/\overline{A_i^{O\Delta}})$, $(CSF_{\overline{A_i^{O\Delta}}}(E)$, $CAF_{\overline{A_i^{O\Delta}}}(E)$;
8:　　　　　　according to Definition 19 to get $A_i^{O\prime}$;
9:　　　　　　**return** $A_i^{O\prime} = A_i$;
10:　　　**end**
11:　　**else**
12:　　　　**return** NULL;
13:　　**end**
14:　**end**
15: **end**
16: **else**
17: **return** $A_i^{O\prime} = A_i^O$;
18: **end**

---

## 6. Example Analysis 3 (Continued with Example 2)

In Example 1, only site 1 can be ignored under optimistic and pessimistic multi-granulation conditions, so it can be determined that site 1 does not need to be evaluated, while sites 2 and 3 need to be further investigated under the environment of optimistic multi-granulation. At the same time, with respect to the environment of pessimistic multi-granulation, comprehensive considera- tion site 3 can ignore the assessment and sites 2, 4 and 5 need to be further investigated.

According to Example 1, we can get that the reduction set of OMRS is $\{A_2, A_3\}$, but in the case of IMRS, there are two reduction sets, which are contradictory. Therefore, two reduction sets should be reconsidered simultaneously, so the joint reduction set under IMRS is $\{A_2, A_4, A_5\}$.

Where the corresponding granularity structures of sites 2, 3, 4 and 5 are divided as follows:

$U/A_2 = \{\{x_1, x_2, x_4\}, \{x_3, x_5, x_7\}, \{x_6, x_8, x_9\}, \{x_{10}\}\}$,
$U/A_3 = \{\{x_1, x_4, x_6\}, \{x_2, x_3, x_5\}, \{x_8\}, \{x_7, x_9, x_{10}\}\}$,
$U/A_4 = \{\{x_1, x_2, x_3, x_5\}, \{x_4\}, \{x_6, x_7, x_8\}, \{x_9, x_{10}\}\}$,
$U/A_5 = \{\{x_1, x_3, x_4, x_6\}, \{x_2, x_7\}, \{x_5, x_8\}, \{x_9, x_{10}\}\}$.

According to reference [11], we can get:
$\alpha = \frac{8-2}{(8-2)+(2-0)} = 0.75$; $\beta = \frac{2-0}{(2-0)+(6-2)} = 0.33$.
The optimal site selection process under optimistic and IMRS is as follows:

(1)　Optimal site selection based on OOMRIFS

According to the Example 2, we can get the values of evaluation functions $\underline{\mu^{OO}}(x_j)$, $(1-\underline{\pi^{OO}}(x_j)) \cdot \alpha$, $(1-\underline{\pi^{OO}}(x_j)) \cdot \beta$, $\overline{\mu^{OO}}(x_j)$, $(1-\overline{\pi^{OO}}(x_j)) \cdot \alpha$ and $(1-\overline{\pi^{OO}}(x_j)) \cdot \beta$ of OOMRIFS, as shown in Table 4.

**Table 4.** The values of evaluation functions for OOMRIFS.

|  | $\underline{\mu^{OO}}(x_j)$ | $(1-\underline{\pi^{OO}}(x_j)) \cdot \alpha$ | $(1-\underline{\pi^{OO}}(x_j)) \cdot \beta$ | $\overline{\mu^{OO}}(x_j)$ | $(1-\overline{\pi^{OO}}(x_j)) \cdot \alpha$ | $(1-\overline{\pi^{OO}}(x_j)) \cdot \beta$ |
|---|---|---|---|---|---|---|
| $x_1$ | 0.25 | 0.63 | 0.2772 | 0.51 | 0.5925 | 0.2607 |
| $x_2$ | 0.49 | 0.6525 | 0.2871 | 0.51 | 0.5925 | 0.2607 |
| $x_3$ | 0.49 | 0.6525 | 0.2871 | 0.54 | 0.6675 | 0.2937 |
| $x_4$ | 0.25 | 0.63 | 0.2772 | 0.51 | 0.5925 | 0.2607 |
| $x_5$ | 0.49 | 0.6525 | 0.2871 | 0.54 | 0.6675 | 0.2937 |
| $x_6$ | 0.25 | 0.5325 | 0.2343 | 0.92 | 0.72 | 0.3168 |
| $x_7$ | 0.09 | 0.7125 | 0.3135 | 0.54 | 0.6675 | 0.2937 |
| $x_8$ | 0.15 | 0.4575 | 0.2013 | 0.15 | 0.4575 | 0.2013 |
| $x_9$ | 0.15 | 0.4575 | 0.2013 | 0.72 | 0.63 | 0.2772 |
| $x_{10}$ | 0.67 | 0.675 | 0.297 | 0.67 | 0.675 | 0.297 |

We can get decision results of the lower and upper approximations of OOMRIFS by three-way decisions of the Section 5.1, as follows:

$POS(\underline{X^{OO}}) = \phi$,
$NEG(\underline{X^{OO}}) = \{x_1, x_4, x_7, x_8, x_9\}$,
$BND(\underline{X^{OO}}) = \{x_2, x_3, x_5, x_6, x_{10}\}$;
$POS(\overline{X^{OO}}) = \{x_6, x_9\}$,
$NEG(\overline{X^{OO}}) = \{x_8\}$,
$BND(\overline{X^{OO}}) = \{x_2, x_3, x_5\}$.

In the light of three-way decisions rules based on OOMRIFS, after getting rid of the objects in the rejection domain, we choose to fuse the objects in the delay domain with those in the acceptance domain for the optimal granularity selection. Therefore, the new granularities $A_2$, $A_3$ are as follows:

$U/\underline{A_2^{OI}} = \{\{x_2\}, \{x_3, x_5\}, \{x_6\}, \{x_{10}\}\}$,
$U/\underline{A_3^{OI}} = \{\{x_2, x_3, x_5\}, \{x_6\}, \{x_{10}\}\}$;
$U/\overline{A_2^{OI}} = \{\{x_1, x_2, x_4\}, \{x_3, x_5, x_7\}, \{x_6, x_9\}, \{x_{10}\}\}$,
$U/\overline{A_3^{OI}} = \{\{x_1, x_4, x_6\}, \{x_2, x_3, x_5\}, \{x_7, x_9, x_{10}\}\}$.

Then, according to Definition 18, we can get:

$$
\begin{aligned}
CSF_{\underline{A_2^{OO}}}(E) &= \frac{1}{s} \times \sum_{j=1, n\in[x_j]_{A_i}}^{n} \frac{|\mu_{A_i}(x_j) - \nu_{A_i}(x_j)|}{|[x_j]_{A_i}|} \\
&= \frac{1}{4} \times \sum_{j=1, n\in[x_j]_{\underline{A_2^{OO}}}}^{10} \frac{|\mu_{\underline{A_2^{OO}}}(x_j) - \nu_{\underline{A_2^{OO}}}(x_j)|}{|[x_j]_{\underline{A_2^{OO}}}|} \\
&= \frac{1}{4} \times \left((0.49 - 0.38) + \frac{(0.49-0.38)+(0.49-0.38)}{2} + (0.25 - 0.46) + (0.67 - 0.23)\right) \\
&= 0.1125,
\end{aligned}
$$

$$
\begin{aligned}
CSF_{\underline{A_3^{OO}}}(E) &= \frac{1}{s} \times \sum_{j=1, n\in[x_j]_{A_i}}^{n} \frac{|\mu_{A_i}(x_j) - \nu_{A_i}(x_j)|}{|[x_j]_{A_i}|} \\
&= \frac{1}{3} \times \sum_{j=1, n\in[x_j]_{\underline{A_3^{OO}}}}^{10} \frac{|\mu_{\underline{A_3^{OO}}}(x_j) - \nu_{\underline{A_3^{OO}}}(x_j)|}{|[x_j]_{\underline{A_3^{OO}}}|} \\
&= \frac{1}{3} \times \left((0.25 - 0.46) + \frac{(0.49-0.38)+(0.49-0.38)+(0.49-0.38)}{3} + (0.81 - 0.14)\right) \\
&= 0.1133;
\end{aligned}
$$

Similarly, we have:

$CSF_{\overline{A_2^{OO}}}(E) = 0.4$, $CSF_{\overline{A_3^{OO}}}(E) = 0.3533$.

　　From the above results, in OOMRIFS, we can see that we can't get the selection result of sites 2 and 3 only according to the comprehensive score function of granularities $A_2$ and $A_3$. Therefore, we need to further calculate the comprehensive accuracies to get the results as follows:

$$\begin{aligned}
CAF_{\underline{A_2^{OO}}}(E) &= \tfrac{1}{s} \times \sum_{j=1, n \in [x_j]_{A_i}}^{n} \frac{|\mu_{A_i}(x_j) + \nu_{A_i}(x_j)|}{|[x_j]_{A_i}|} \\
&= \tfrac{1}{4} \times \sum_{j=1, n \in [x_j]_{\underline{A_2^{OO}}}}^{10} \frac{|\mu_{\underline{A_2^{OO}}}(x_j) + \nu_{\underline{A_2^{OO}}}(x_j)|}{|[x_j]_{\underline{A_2^{OO}}}|} \\
&= \tfrac{1}{4} \times \left( (0.49 + 0.38) + \tfrac{(0.49+0.38)+(0.49+0.38)}{2} + (0.25 + 0.46) + (0.67 + 0.23) \right) \\
&= 0.8375,
\end{aligned}$$

$$\begin{aligned}
CAF_{\underline{A_3^{OO}}}(E) &= \tfrac{1}{s} \times \sum_{j=1, n \in [x_j]_{A_i}}^{n} \frac{|\mu_{A_i}(x_j) + \nu_{A_i}(x_j)|}{|[x_j]_{A_i}|} \\
&= \tfrac{1}{3} \times \sum_{j=1, n \in [x_j]_{\underline{A_3^{OO}}}}^{10} \frac{|\mu_{\underline{A_3^{OO}}}(x_j) + \nu_{\underline{A_3^{OO}}}(x_j)|}{|[x_j]_{\underline{A_3^{OO}}}|} \\
&= \tfrac{1}{3} \times \left( (0.25 + 0.46) + \tfrac{(0.49+0.38)+(0.49+0.38)+(0.49+0.38)}{3} + (0.81 + 0.14) \right) \\
&= 0.8267;
\end{aligned}$$

Analogously, we have:
$CAF_{\overline{A_2^{OO}}}(E) = 0.87, CAF_{\overline{A_3^{OO}}}(E) = 0.86.$

　　Through calculation above, we know that the comprehensive accuracy of the granularity $A_3$ is higher, so the site 3 is selected as the selection result.

(2)　Optimal site selection based on OIMRIFS

　　The same as (1), we can get the values of evaluation functions $\underline{\mu^{OI}}(x_j)$, $(1 - \underline{\pi^{OI}}(x_j)) \cdot \alpha$, $(1 - \underline{\pi^{OI}}(x_j)) \cdot \beta$, $\overline{\mu^{OI}}(x_j)$, $(1 - \overline{\pi^{OI}}(x_j)) \cdot \alpha$ and $(1 - \overline{\pi^{OI}}(x_j)) \cdot \beta$ of OIMRIFS listed in Table 5.

**Table 5.** The values of evaluation functions for OIMRIFS.

| | $\underline{\mu^{OI}}(x_j)$ | $(1-\underline{\pi^{OI}}(x_j))\cdot\alpha$ | $(1-\underline{\pi^{OI}}(x_j))\cdot\beta$ | $\overline{\mu^{OI}}(x_j)$ | $(1-\overline{\pi^{OI}}(x_j))\cdot\alpha$ | $(1-\overline{\pi^{OI}}(x_j))\cdot\beta$ |
|---|---|---|---|---|---|---|
| $x_1$ | 0.25 | 0.63 | 0.2772 | 0.92 | 0.72 | 0.3168 |
| $x_2$ | 0.25 | 0.63 | 0.2772 | 0.54 | 0.615 | 0.2706 |
| $x_3$ | 0.09 | 0.7125 | 0.3135 | 0.54 | 0.615 | 0.2706 |
| $x_4$ | 0.25 | 0.63 | 0.2772 | 0.92 | 0.72 | 0.3168 |
| $x_5$ | 0.09 | 0.7125 | 0.3135 | 0.54 | 0.615 | 0.2706 |
| $x_6$ | 0.15 | 0.555 | 0.2442 | 0.92 | 0.72 | 0.3168 |
| $x_7$ | 0.09 | 0.7125 | 0.3135 | 0.72 | 0.63 | 0.2772 |
| $x_8$ | 0.15 | 0.4575 | 0.2013 | 0.92 | 0.72 | 0.3168 |
| $x_9$ | 0.09 | 0.7125 | 0.3135 | 0.92 | 0.72 | 0.3168 |
| $x_{10}$ | 0.09 | 0.7125 | 0.3135 | 0.72 | 0.63 | 0.2772 |

　　We can get decision results of the lower and upper approximations of OIMRIFS by three-way decisions in the Section 5.2, as follows:
$POS(\underline{X^{OI}}) = \phi,$
$NEG(\underline{X^{OI}}) = U,$
$BND(\underline{X^{OI}}) = \phi;$
$POS(\overline{X^{OI}}) = \{x_1, x_4, x_6, x_7, x_8, x_9, x_{10}\},$
$NEG(\overline{X^{OI}}) = \phi,$
$BND(\overline{X^{OI}}) = \{x_2, x_3, x_5\}.$
　　Hence, in the upper approximations of OIMRIFS, the new granularities $A_2$, $A_3$ are as follows:

$U/\underline{A_2^{OI}} = \{\{x_1, x_2, x_4\}, \{x_3, x_5, x_7\}, \{x_6, x_8, x_9\}, \{x_{10}\}\},$

$U/\underline{A_3^{OI}} = \{\{x_1, x_4, x_6\}, \{x_2, x_3, x_5\}, \{x_8\}, \{x_7, x_9, x_{10}\}\}.$

According to Definition 18, we can calculate that

$CSF_{\underline{A_2^{OI}}}(E) = CSF_{\underline{A_3^{OI}}}(E) = 0;$

$CAF_{\underline{A_2^{OI}}}(E) = CAF_{\underline{A_3^{OI}}}(E) = 0;$

$CSF_{\overline{A_2^{OI}}}(E) = 0.6317, CSF_{\overline{A_3^{OI}}}(E) = 0.6783;$

$CAF_{\overline{A_2^{OI}}}(E) = 0.885, CAF_{\overline{A_3^{OI}}}(E) = 0.905.$

In OIMRIFS, the comprehensive score and comprehensive accuracy of the granularity $A_3$ are both higher than the granularity $A_2$. So, we choose site 3 as the evaluation site.

In reality, we are more inclined to select the optimal granularity in the case of more stringent requirements. According to (1) and (2), we can find that the granularity $A_3$ is a better choice when the requirements are stricter in four cases of OMRS. Therefore, we choose site 3 as the optimal evaluation site.

(3) Optimal site selection based on IOMRIFS

Similar to (1), we can obtain the values of evaluation functions $\underline{\mu^{IO}}(x_j)$, $(1 - \underline{\pi^{IO}}(x_j)) \cdot \alpha$, $(1 - \underline{\pi^{IO}}(x_j)) \cdot \beta$, $\overline{\mu^{IO}}(x_j)$, $(1 - \overline{\pi^{IO}}(x_j)) \cdot \alpha$ and $(1 - \overline{\pi^{IO}}(x_j)) \cdot \beta$ of IOMRIFS, as described in Table 6.

**Table 6.** The values of evaluation functions for IOMRIFS.

| | $\underline{\mu^{IO}}(x_j)$ | $(1-\underline{\pi^{IO}}(x_j))\cdot\alpha$ | $(1-\underline{\pi^{IO}}(x_j))\cdot\beta$ | $\overline{\mu^{IO}}(x_j)$ | $(1-\overline{\pi^{IO}}(x_j))\cdot\alpha$ | $(1-\overline{\pi^{IO}}(x_j))\cdot\beta$ |
|---|---|---|---|---|---|---|
| $x_1$ | 0.25 | 0.51 | 0.2244 | 0.51 | 0.5925 | 0.2607 |
| $x_2$ | 0.25 | 0.51 | 0.2244 | 0.51 | 0.5925 | 0.2607 |
| $x_3$ | 0.25 | 0.51 | 0.2244 | 0.54 | 0.6675 | 0.2937 |
| $x_4$ | 0.37 | 0.72 | 0.3168 | 0.37 | 0.72 | 0.3168 |
| $x_5$ | 0.25 | 0.51 | 0.2244 | 0.49 | 0.63 | 0.2772 |
| $x_6$ | 0.25 | 0.5325 | 0.2343 | 0.92 | 0.72 | 0.3168 |
| $x_7$ | 0.09 | 0.7125 | 0.3135 | 0.51 | 0.645 | 0.2838 |
| $x_8$ | 0.15 | 0.4575 | 0.2013 | 0.49 | 0.63 | 0.2772 |
| $x_9$ | 0.67 | 0.675 | 0.297 | 0.72 | 0.63 | 0.2772 |
| $x_{10}$ | 0.67 | 0.675 | 0.297 | 0.67 | 0.675 | 0.297 |

We can get decision results of the lower and upper approximations of IOMRIFS by three-way decisions in the Section 5.3, as follows:

$POS(\underline{X^{IO}}) = \phi,$

$NEG(\underline{X^{IO}}) = \{x_7, x_8\},$

$BND(\underline{X^{IO}}) = \{x_1, x_2, x_3, x_4, x_5, x_6, x_9, x_{10}\};$

$POS(\overline{X^{IO}}) = \{x_6, x_9\},$

$NEG(\overline{X^{IO}}) = \phi,$

$BND(\overline{X^{IO}}) = \{x_1, x_2, x_3, x_4, x_5, x_7, x_8, x_{10}\}.$

Therefore, the granularities $A_2$, $A_4$, $A_5$ can be rewritten as follows:

$U/\underline{A_2^{IO}} = \{\{x_1, x_2, x_4\}, \{x_3, x_5\}, \{x_6, x_9\}, \{x_{10}\}\},$

$U/\underline{A_4^{IO}} = \{\{x_1, x_2, x_3, x_5\}, \{x_4\}, \{x_6\}, \{x_9, x_{10}\}\},$

$U/\underline{A_5^{IO}} = \{\{x_1, x_3, x_4, x_6\}, \{x_2\}, \{x_5\}, \{x_9, x_{10}\}\};$

$U/\overline{A_2^{IO}} = \{\{x_1, x_2, x_4\}, \{x_3, x_5, x_7\}, \{x_6, x_8, x_9\}, \{x_{10}\}\},$

$U/\overline{A_4^{IO}} = \{\{x_1, x_2, x_3, x_5\}, \{x_4\}, \{x_6, x_7, x_8\}, \{x_9, x_{10}\}\},$

$U/\overline{A_5^{IO}} = \{\{x_1, x_3, x_4, x_6\}, \{x_2, x_7\}, \{x_5, x_8\}, \{x_9, x_{10}\}\}.$

According to Definition 18, one can see that the results are captured as follows:

$CSF_{\underline{A_2^{IO}}}(E) = 0.0454, CSF_{\underline{A_4^{IO}}}(E) = -0.0567, CSF_{\underline{A_5^{IO}}}(E) = -0.0294;$

$CSF_{\overline{A_2^{IO}}}(E) = 0.3058, CSF_{\overline{A_4^{IO}}}(E) = 0.2227, CSF_{\overline{A_5^{IO}}}(E) = 0.2813.$

In summary, the comprehensive score function of the granularity $A_2$ is higher than the granularity $A_3$ in IOMRIFS, so we choose site 2 as the result of granularity selection.

(4)　Optimal site selection based on IIMRIFS

In the same way as (1), we can get the values of evaluation functions $\underline{\mu}^{II}(x_j)$, $(1 - \underline{\pi}^{II}(x_j)) \cdot \alpha$, $(1 - \underline{\pi}^{II}(x_j)) \cdot \beta$, $\overline{\mu^{II}}(x_j)$, $(1 - \overline{\pi^{II}}(x_j)) \cdot \alpha$ and $(1 - \overline{\pi^{II}}(x_j)) \cdot \beta$ of IIMRIFS, as shown in Table 7.

**Table 7.** The values of evaluation functions for IIMRIFS.

|  | $\underline{\mu}^{II}(x_j)$ | $(1-\underline{\pi}^{II}(x_j))\cdot\alpha$ | $(1-\underline{\pi}^{II}(x_j))\cdot\beta$ | $\overline{\mu^{II}}(x_j)$ | $(1-\overline{\pi^{II}}(x_j))\cdot\alpha$ | $(1-\overline{\pi^{II}}(x_j))\cdot\beta$ |
|---|---|---|---|---|---|---|
| $x_1$ | 0.25 | 0.63 | 0.2772 | 0.92 | 0.72 | 0.3168 |
| $x_2$ | 0.09 | 0.7125 | 0.3135 | 0.54 | 0.615 | 0.2706 |
| $x_3$ | 0.09 | 0.7125 | 0.3135 | 0.92 | 0.72 | 0.3168 |
| $x_4$ | 0.25 | 0.63 | 0.2772 | 0.92 | 0.72 | 0.3168 |
| $x_5$ | 0.09 | 0.7125 | 0.3135 | 0.54 | 0.615 | 0.2706 |
| $x_6$ | 0.09 | 0.7125 | 0.3135 | 0.92 | 0.72 | 0.3168 |
| $x_7$ | 0.09 | 0.7125 | 0.3135 | 0.92 | 0.72 | 0.3168 |
| $x_8$ | 0.09 | 0.7125 | 0.3135 | 0.92 | 0.72 | 0.3168 |
| $x_9$ | 0.15 | 0.4575 | 0.2013 | 0.92 | 0.72 | 0.3168 |
| $x_{10}$ | 0.67 | 0.675 | 0.297 | 0.72 | 0.63 | 0.2772 |

We can get decision results of the lower and upper approximations of IIMRIFS by three-way decisions in the Section 5.4, as follows:

$POS(\underline{X^{II}}) = \phi$,

$NEG(\underline{X^{II}}) = \{x_1, x_2, x_3, x_4, x_5, x_6, x_7, x_8, x_9\}$,

$BND(\underline{X^{II}}) = \{x_{10}\}$;

$POS(\overline{X^{II}}) = \{x_1, x_3, x_4, x_6, x_7, x_8, x_9, x_{10}\}$,

$NEG(\overline{X^{II}}) = \phi$,

$BND(\overline{X^{II}}) = \{x_2, x_5\}$.

Therefore, the granularity structures of $A_2$, $A_4$, $A_5$ can be rewritten as follows:

$U/\underline{A_2^{II}} = U/\underline{A_4^{II}} = U/\underline{A_5^{II}} = \{x_{10}\}$;

$U/\overline{A_2^{II}} = \{\{x_1, x_2, x_4\}, \{x_3, x_5, x_7\}, \{x_6, x_8, x_9\}, \{x_{10}\}\}$,

$U/\overline{A_4^{II}} = \{\{x_1, x_2, x_3, x_5\}, \{x_4\}, \{x_6, x_7, x_8\}, \{x_9, x_{10}\}\}$,

$U/\overline{A_5^{II}} = \{\{x_1, x_3, x_4, x_6\}, \{x_2, x_7\}, \{x_5, x_8\}, \{x_9, x_{10}\}\}$.

According to Definition 18, one can see that the results are captured as follows:

$CSF_{\underline{A_2^{II}}}(E) = CSF_{\underline{A_4^{II}}}(E) = CSF_{\underline{A_5^{II}}}(E) = 0.44$;

$CAF_{\underline{A_2^{II}}}(E) = CAF_{\underline{A_4^{II}}}(E) = CAF_{\underline{A_5^{II}}}(E) = 0.9$;

$CSF_{\overline{A_2^{II}}}(E) = 0.7067$, $CSF_{\overline{A_4^{II}}}(E) = 0.7675$, $CSF_{\overline{A_5^{II}}}(E) = 0.69$;

$CAF_{\overline{A_2^{II}}}(E) = 0.9067$, $CAF_{\overline{A_4^{II}}}(E) = 0.9275$, $CAF_{\overline{A_5^{II}}}(E) = 0.91$.

In IIMRIFS, the values of the comprehensive score and comprehensive accuracy of granularity $A_4$ are higher than $A_2$ and $A_5$, so site 4 is chosen as the evaluation site.

Considering (3) and (4) synthetically, we find that the results of granularity selection in IOMRIFS and IIMRIFS are inconsistent, so we need to further compute the comprehensive accuracies of IIMRIFS.

$CAF_{\underline{A_2^{IO}}}(E) = 0.7896$, $CAF_{\underline{A_4^{IO}}}(E) = 0.8125$, $CAF_{\underline{A_5^{IO}}}(E) = 0.7544$;

$CAF_{\overline{A_2^{IO}}}(E) = 0.8725$, $CAF_{\overline{A_4^{IO}}}(E) = 0.886$, $CAF_{\overline{A_5^{IO}}}(E) = 0.8588$.

Through the above calculation results, we can see that the comprehensive score and comprehensive accuracy of granularity $A_4$ are higher than $A_2$ and $A_5$ in the case of pessimistic multi- granulation when the requirements are stricter. Therefore, the site 4 is eventually chosen as the optimal evaluation site.

## 7. Conclusions

In this paper, we propose two new granularity importance degree calculating methods among multiple granularities, and a granularity reduction algorithm is further developed. Subsequently, we design four novel MRIFS models based on reduction sets under optimistic and IMRS, i.e., OOMRIFS, OIMRIFS, IOMRIFS, and IIMRIFS, and further demonstrate their relevant properties. In addition, four three-way decisions models with novel MRIFS for the issue of internal redundant objects in reduction sets are constructed. Finally, we designe the comprehensive score function and the comprehensive precision function for the optimal granularity selection results. Meanwhile, the validity of the proposed models is verified by algorithms and examples. The works of this paper expand the application scopes of MRIFS and three-way decisions theory, which can solve issues such as spam e-mail filtering, risk decision, investment decisions, and so on. A question worth considering is how to extend the methods of this article to fit the big data environment. Moreover, how to combine the fuzzy methods based on triangular or trapezoidal fuzzy numbers with the methods proposed in this paper is also a research problem. These issues will be investigated in our future work.

**Author Contributions:** Z.-A.X. and D.-J.H. initiated the research and wrote the paper, M.-J.L. participated in some of these search work, and M.Z. supervised the research work and provided helpful suggestions.

**Funding:** This research received no external funding.

**Acknowledgments:** This work is supported by the National Natural Science Foundation of China under Grant Nos. 61772176, 61402153, and the Scientific And Technological Project of Henan Province of China under Grant Nos. 182102210078, 182102210362, and the Plan for Scientific Innovation of Henan Province of China under Grant No. 18410051003, and the Key Scientific And Technological Project of Xinxiang City of China under Grant No. CXGG17002.

**Conflicts of Interest:** The authors declare no conflicts of interest.

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
