# Peer review of "Novel Three-Way Decisions Models with Multi-Granulation Rough Intuitionistic Fuzzy Sets"

_symmetry, doi:10.3390/sym10110662_

Reviewer 1 Report

- The proposal is appealing and interesting, and the method deserves some consideration. Moreover, the paper is almost well written and well organized.

- The text, in general, reads well, but a grammatical revision could improve it further.

- The paper almost puts the progress it reports in the context of previous works, representative referencing and first discussion. However, the analysis of further recent works in literature could be included.

- The authors could highlight better the new scientific contribution.

Author Response

Thank you for your letter and for the reviewers’ comments concerning our manuscript entitled “Novel three-way decisions models with multi-granulation rough intuitionistic fuzzy sets” (ID: symmetry-387534). Those comments are all valuable and very helpful for revising and improving our paper, as well as the important guiding significance to our researches. We have studied comments carefully and have made correction which we hope meet with approval. The content modified was marked in blue color in revised paper. The point to point responds to the reviewer’s comments are listed as file "Response to Reviewer 1 Comments".

Reviewer 2 Report

The article is based on a correct mathematical apparatus, and contains new and significant information adequate to justify publication. However, I propose minor amendments:

In my opinion, the structure of the paper should be modified. The computational example 1 should be placed in a single section before conclusions. Generally, the paper should have a classic structure: introduction, materials and methods (i.a. all theoretical definitions, theorems, proofs and algorithms), results (application of the method – example 1), discussion and conclusion.

The Conclusion section ought to be improved. The section should additionally discuss the potential real applications of the proposed method and directions of the further research on the method. Moreover, the advantages of the method in the comparison to the fuzzy methods based on triangular or trapeziodal fuzzy numbers should be clearly pointed out.

Author Response

Thank you for your letter and for the reviewers’ comments concerning our manuscript entitled “Novel three-way decisions models with multi-granulation rough intuitionistic fuzzy sets” (ID: symmetry-387534). Those comments are all valuable and very helpful for revising and improving our paper, as well as the important guiding significance to our researches. We have studied comments carefully and have made correction which we hope meet with approval. The content modified was marked in blue color in revised paper. The point to point responds to the reviewer’s comments are listed as file "Response to Reviewer 2 Comments".

Round  2

Reviewer 1 Report

The authors have addressed my comments. As a consequence, the paper can be accepted.